# Regional Differences in Ecological Compensation for Cultivated Land Protection: An Analysis of Chengdu, Sichuan Province, China

**DOI:** 10.3390/ijerph17218242

**Published:** 2020-11-08

**Authors:** Kunpeng Wang, Minghao Ou, Zinabu Wolde

**Affiliations:** 1College of Land Management, Nanjing Agricultural University, Nanjing 210095, China; 2018209019@njau.edu.cn (K.W.); sos.zine04@gmail.com (Z.W.); 2Center of Urban–Rural Joint Development and Land Management Innovation, Nanjing 210095, China

**Keywords:** cultivated land protection, ecological compensation, Profitable Spatial Boundary Model (PSBM), external environmental differences, China

## Abstract

Exploring the elements that affect farmers’ willingness to protect cultivated land is the key to improving the ecological compensation mechanism for cultivated land protection. The purpose of this study was to analyze regional differences in ecological compensation for cultivated land protection, and to explore the influence of different external environments on farmers’ willingness to engage in cultivated land protection. Based on the Profitable Spatial Boundary Analysis theory (PSBA), GIS spatial analysis technology was used to analyze regional space differences and assess ecological compensation for urban and rural cultivated land protection at the micro scale. The results show that the willingness of farmers to participate in cultivated land protection is affected by the external environment and the ecological compensation offered. The trend of the comprehensive benefit of cultivated land protection ecological compensation (*B*) is “Λ” from the first layer to the third layer. The *B* value of the urban–rural junction area is the highest value. This shows that the external environment is favorable for ecological compensation in this area, which has a positive effect on farmers’ willingness to protect cultivated land. *B* < 0 in the first and third layer, which has a depressant effect on farmers’ willingness to protect cultivated land. The study results contribute to the understanding of the impact of regional differences in the external environmental on ecological compensation and farmers’ willingness to engage in cultivated land protection.

## 1. Introduction

In developing countries, there is a significant income gap between nonagricultural employment and agricultural employment [1]. The high income from nonagricultural employment attracts many agricultural laborers to work in the city [1,2]. Thus, the cultivated land is poorly protected, and is abandoned, which results in food security being a threat in developing countries [3]. In addition, the destruction of cultivated land ecosystems will negatively affect the ecological environment needed for human survival as well. Establishing an ecological compensation mechanism for cultivated land protection is an important measure to achieve coordinated development of the social economy and the environment [2,3]. Due to the large number of stakeholders and the long implementation period involved in developing such a mechanism, it is difficult to implement the compensation policy for cultivated land protection in developing countries. Any issues that arise in the implementation process are closely related to the implementation effect [4], which could affect a government’s evaluation of this policy. Thus, it is necessary to be cautious regarding the implementation of an ecological compensation policy for cultivated land protection. International scholars initially focused on the concept of “natural service” [5], the theoretical basis [6] and classification of ecosystem service value [7]. With a deeper understanding of ecosystem services, scholars have gradually paid more attention to the relationship between land use change and impact on ecosystem services [8,9]. In the present study, the research on cultivated land mainly focused on cultivated land development rights [10,11,12], payment willingness [13], factors influencing cultivated land protection [14,15], the value of cultivated land [16], and ecological benefit evaluation [17,18]. In China, scholars have mainly focused on cultivated land protection compensation [19], compensation methods [20], standards of compensation [21], willingness to provide compensation [22], and the construction of compensation mechanisms [19,23]. Among these topics, research on willingness to provide ecological compensation has attracted much attention [24]. The degree of willingness of farmers is the main factor influencing policy on ecological compensation for cultivated land protection [19].

Farmers are a core shareholder in protecting cultivated land. The purpose of ecological compensation for cultivated land protection is to stimulate farmers’ enthusiasm for cultivated land protection, increase their income, and improve their livelihood and security [25]. Meanwhile, farmers’ willingness and behaviors are not only influenced by individual and family characteristics, but also by their external environment (e.g., the natural condition of cultivated land, protection policy of cultivated land, and regional environment) [26,27]. Due to geographical differences, as well as the economic development level and employment opportunities, different compensation modes are formed. The “Chengdu mode” and “Shanghai mode” are based on the division of basic farmland and general farmland. The “Foshan mode” and “Guangzhou mode” are compensation modes distinguished by the geographical location of the farmland. The “Suzhou mode” and “Haining mode” are compensation modes based on the cultivated land scale [19]. From the above discussion, this study aims to investigate the internal mechanism that affects the willingness of farmers to protect cultivated land in terms of the external environmental differences by regions. This will allow us to understand how external environmental differences affect farmers’ willingness to engage in cultivated land protection. These issues are still not clarified.

## 2. Theoretical Analysis

### 2.1. External Environment of Ecological Compensation for Cultivated Land Protection

There are stark differences between urban and rural areas in China [1,2,3,27]. These differences mainly include the degree of urban economic development [20], income level [21], employment environment [22], social security [23], medical benefits [24]. According to the “Economic Man Hypothesis (EMH)”, people pursue their own maximized interests [25,28,29]. However, due to the huge population, limited cultivated land resources, and strict cultivated land protection policies in China, it is impossible for farmers to fully pursue the goal of maximizing economic benefits [1]. To achieve the expected benefits, farmers will make a comprehensive judgment on the external environment of the ecological compensation of cultivated land protection [27]. In general, farmers’ cultivated land protection adheres to the concept of “bounded rationality”.

The Bounded Rationality Theory (BRT) illustrates the process of decision making and indicates various elements of decision making. The rationality of decision-making under BRT is not the absolute optimal solution, but is the most satisfactory solution among all alternative solutions [30,31,32]. According to BRT, the decision-making object is influenced by its own cognitive factors, external environmental factors, and random interference terms. The given expression further clarifies the composition factors of BRT [30,31]:(1)Q=L(c,e,i)

In Equation (1), *c* represents the cognitive factors of the decision-making subject, *e* represents the external environment factors, and *i* represents the random interference terms. Therefore, analyzing the external environment is conducive to improving the ecological compensation system.

Based on the distance from the city center, the geographical location of farmers can be divided into urban villages, rural areas near the urban areas, and remote rural areas. From the perspective of spatial location, the regional differences can be divided into a central urban area, urban–rural junction area, and typical rural area. In the process of urban–rural integration, there are differences in regional economic development, land use status [3], land circulation degree [4], cultivated land protection policies [2], social security [17], and industry types in different spatial locations [18] (Figure 1). These differences may be the key drivers of the external environment of cultivated land protection ecological compensation under different conditions.

Generally speaking, the higher the level of regional economic development in the area, the closer it is to a city. The central urban area and urban–rural junction area are close to cities and have many industries. However, the closer the farmers are to the town, the more employment opportunities they have, the less dependent they are on land, and the higher the degree of land circulation [33]. On the other hand, the economic development level of the typical rural area is relatively low. In fact, due to farmers’ low level of knowledge and technology, they have fewer opportunities to work in cities, and they rely more on the land [17,34]. Therefore, farmers in the typical rural area have a low willingness to transfer land. Moreover, policy is the key factor that prevents cultivated land loss and fragmentation [26]. Some supervision measures are more conducive to the implementation of cultivated land protection policies.

Farmers are the main participants in the implementation of ecological compensation policies for cultivated land protection [35]. The external environment of farmers’ cultivated land protection is different from their own internal factors [36]. The external environment of cultivated land protection for farmers is affected by market fluctuations and policies. Therefore, this external influence can be improved through measures taken by the government.

### 2.2. External Environment Factors of Cultivated Land Ecological Compensation

Cultivated land is the absolute foundation of human beings’ survival today and in the future and, as such, it must be protected from conversion into built-up environments and fragmentation. With the largest population in the world and rapid urbanization, China has a particular need to protect its cultivated land [2]. Since the start of the reforms and openness policy in 1978, the urbanization of China has been accompanied by a series of large-scale land use changes [20]. Meanwhile, there are stark differences between urban and rural areas in China [1,2,3,27]. From the perspective of geographical location, farmers can be divided into urban villages, rural areas near the urban areas, and remote rural areas. Based on the BRT, the willingness of farmers to undertake cultivated land protection is affected by the external environment [36]. The yield of cultivated land is closely related to the factors agricultural production environment. The improvement of agricultural production environment contributes to the increase of cultivated land yield. However, the regional agricultural production environment depends on the cultivated land protection policy environment [22]. To the farmers, the degree of protection for cultivated land is influenced by the different factors of the living environments. In addition, the employment environment affects the number of farmers working in agriculture. Furthermore, in China, cultivated land serves as both a means of production and as social insurance for farmers. The agricultural subsidy and ecological compensation for cultivated land can directly increase farmers’ income and consumption level [17]. The representative variables were analyzed in this study. The variables affecting ecological compensation for farmers’ cultivated land protection were defined from the perspective of the agricultural production environment, farmers’ living environment, farmers’ employment environment, farmers’ safeguarding environment, the market environment, and policy environment of the ecological compensation for cultivated land protection (Figure 2).

Because the main function of cultivated land is to provide food production [37], in this study, the total agricultural output value was an indicator representing the agricultural production status. The higher the regional agricultural output, the more advantageous the regional agricultural production conditions are [24]. The stable development of agriculture requires sufficient agricultural labor force [38]. Therefore, as indicators to reflect the production environment of ecological compensation for cultivated land protection, the total agricultural output value and population machinery growth rate were chosen.

Moreover, the more the number of regional industries, the more attractive it is for rural households to work in cities. A large number of rural laborers migrate to cities and towns, which indirectly reduces the direct implementation groups of cultivated land protection [33]. In terms of budget expenditure, as it lacks a thorough evaluation mechanism, it is difficult to match the public budget expenditure with social income, or the net benefits may even be negative [39]. However, blindly expanding the scale of budget expenditure will reduce social productivity. Therefore, service industries above a designated size and general public budget expenditure are selected to reflect the living environment of the study area.

In general, population density and the average number of employees are positively correlated with regional employment pressure. Once the degree of competition increases, it will cause greater psychological pressure on farmers with weak competitiveness [26]. To maintain the current living standard, farmers are unwilling to increase the expenditure on cultivated land protection too much [40]. At the same time, the higher a household’s income level is, the more likely farmers are to invest in cultivated land protection [1]. The improvement of the rural social security mechanism is conducive to an increase in farmers’ willingness to protect cultivated land [31]. In 2008, Chengdu was the pilot area for exploring the cultivated land protection ecological compensation system in China. To implement the cultivated land protection ecological compensation policy, Chengdu set up a special fund for farmers who protect cultivated land. This special cultivated land protection fund was for farmers to purchase endowment insurance [2]. The purpose is to provide livelihood security and encourage farmers to protect cultivated land. Farmers’ pension issues are also considered, which is a highlight of the ecological compensation policy in Chengdu. From what has been discussed above, two indicators are selected to reflect the safeguarding environment: the number of rural medical and health institutions and the minimum living security expenditure of urban and rural residents.

In addition, when the household income of rural households remains unchanged, the increase in retail sales of rural consumer goods means that rural households spend more on living expenses, which will reduce the investment in agriculture [39]. At the same time, the increase in the capital stock can promote economic development [41], adjust the industrial structure, and optimize the environment for implementing the policy of cultivated land protection ecological compensation. Therefore, fixed asset investment was selected as an indicator of the policy environment.

### 2.3. Profitable Spatial Boundary Analysis Theory (PSBA)

Farmers evaluate the costs and benefits of their cultivated land protection behaviors based on a comprehensive judgment of the external environment [30,31,32]. For evaluation of this situation, the most suitable theory is the Profitable Spatial Boundary Analysis Theory (PSBA), which was proposed by D.M. Smith [42].

Participants do not always make the best locational choices, because they are affected by the availability of information and subjective factors. Smith’s PSBA emphasizes the importance of human behavior in location selection. Smith combined space cost curve theory by Weber [43] and space revenue curve theory by Losch [44], through space boundary analysis to find the best location [30,31,32].

In Figure 3, *A*_1_ refers to the highest point of the Space revenue curve *P* (*TR*) and *B*_2_ refers to the lowest point of the Space cost curve *P* (*TC*). *A*_1_*A*_2_ refers to the profit value of the *A* location, and *B*_1_*B*_2_ is the profit value of the *B* location. *K_a_* and *K_b_* are two points where space cost and space revenue are equal in geospatial location [42,43,44]. In the *K_a_K_b_* geographic interval, the *P* (*TR*) is greater than the space cost curve *AC* (*TC*). Therefore, the irregular quadrilateral *K_a_*_’_*B*_2_*K_b_*_’_*A*_1_ is the profit zone, and *K_a_* and *K_b_* are the geographical boundaries of the profit.

In Figure 4, under an unchanged space revenue curve, if the cost of cultivated land protection space decreases, the space profit area will be expanded from the original quadrangular area *K_a_*_’_*B*_2_*K_b_*_’_*A*_1_ to the quadrangular area *C*_1_*B*_3_*D*_1_*A*_1_, which indicates an increase in profits. If the cost of space for cultivated land protection increases, then the space profit area is reduced from the original quadrangular area *K_a_*_’_*B*_2_*K_b_*_’_*A*_1_ to the quadrangular area *C*_1_*B*_3_*D*_1_*A*_1_, which indicates that the profit is reduced [42,43,44].

### 2.4. Classification of Factors Based on PSBA

The factors are divided into positive and negative factor categories. Positive external factors include the factors of the production environment, safeguarding environment, and policy environment. If the value of the positive factor is larger, the external environment is favorable for ecological compensation of cultivated land protection in this area. At the same time, farmers will increase their revenue after implementing cultivated land protection, and their willingness to protect cultivated land will increase. Therefore, according to the PSBL, these factors’ attributes are positive, and they are recorded as being revenue factors (*TR*) of ecological compensation for cultivated land protection.

Negative external factors of cultivated land protection ecological compensation include the living environment, employment environment, and market environment. If the values of the positive factors are larger, the external environment is worse for cultivated land protection ecological compensation in this area. At the same time, farmers will decrease their revenue after implementing cultivated land protection, and their willingness to protect cultivated land will decrease. Therefore, these factors’ attributes are negative indicators, and they are recorded as cost factors (*TC*) of cultivated land protection ecological compensation.

### 2.5. Model Specification Based on PSBA

After a comprehensive assessment of the external environment, farmers will consider whether to adopt measures to protect the cultivated land. In particular, for farmers who take measures to protect cultivated land, they are most concerned about how to maximize their profits [33]. Based on the PSBA, under the current external environment of the region, the predictable total revenue and the total cost are the key factors for famers to consider. When the predictable total revenue is greater than the total cost, the comprehensive benefit of external environment of ecological compensation for cultivated land protection is positive. This means that the external environment of the region is conducive to farmers’ increasing willingness to protect cultivated land. However, when the predictable total revenue is less than the total cost, the comprehensive benefit of external environment of ecological compensation for cultivated land protection is negative. This means that the external environment of the region is not conducive to improving farmers’ willingness to protect cultivated land. Therefore, the model specification can be shown as:(2)B = TR−TC
(3)$$\begin{array}{l} TR=\sum_{i=1}^{n}K_{i}P_{i}\\ TC=\sum_{i=1}^{m}K_{i}C_{i} \end{array}$$

Equation (2) is based on the authors’ own illustration drawn from the studied empirical and theoretical literature; however, Equation (3) is based on prior literature [30,43,44]. In Equation (2), *B* represents the comprehensive benefit of external environment of ecological compensation for cultivated land protection, *TR* refers to the predictable total revenue of farmer households under the external environment for the ecological compensation of cultivated land protection, *TC* refers to the predictable total cost of farmer households. Likewise, in Equation (3), *P_i_* represents some positive indicators, *n* is the number of positive indicators, *C_i_* represents negative, and *K_i_* represents the weight of indicators, *m* is the number of negative indicators. Figure 5 shows the PSBL of the external environment for the ecological compensation of cultivated land protection. The *K_a_* and *K_b_* interval shows *TR* > *TC* and *B* > 0, which indicates that the predictable total revenue is greater than the total cost under the current external environment of the region. The external environment of the region is conducive to farmers’ increasing willingness to protect cultivated land.

## 3. Materials and Methods

### 3.1. Study Area

Chengdu is located in southwest China, between 102°54′ E and 104°53′ E and 30°05′ N and 31°26′ N. The land area of the city is 14,335 km^2^, and the cultivated land area is 4320 km^2^ [45]. The city’s terrain slopes from the northwest to the southeast, and it has fertile land and deep soil layers. The proportion of available area is 94.2%, of which the area of the plain area is as high as 60% or more [46]. Plains, hills, and mountains account for 40.1%, 27.6%, and 32.3% of the city’s total land area, respectively [45]. In 2017, the total population of Chengdu was 13.9893 million, with 6.1433 million rural residents and 7.846 million urban residents [46].

There are 21 counties and districts in Chengdu city (including Jianyang City and the High-Tech zone). There are significant differences in terms of the economic and social development levels among different regions, and the development trend of the urban circle is from the city center to the outside. The “first layer” consists of six districts (Jinjiang District, Wuhou District, Qingyang District, Jinniu District, Chenghua District, and the High-Tech District) [46], which are mainly located in the central city and have many high-tech cultural industries. With a good location and the advantages of development, they show a trend of “leading development” in the municipal economy. The “second layer” consists of six suburban districts and counties (Long Quan-Yi District, Qing Baijiang District, Xindu District, Wenjiang District, Shuangliu District, and Lixian County). The conditions are suitable, and secondary industry is mainly located here. There is a trend of “median development” in the municipal economy. The “third layer” includes nine cities and counties (Du Jiangyan City, Pengzhou City, Qiong City, Chongzhou City, Jintang County, Dayi County, Pujiang County, Xinjin County, and Jianyang City), which are far from the city and have a poor location and weak development. The agricultural area is mainly located here. There is a trend of “subsequent development” in the municipal economy. Chengdu has a clear layer structure, which is regarded as representative of many cities with circular development. Therefore, as the empirical analysis area, Chengdu has representative regional differences in the external environment in terms of cultivated land protection ecological compensation [45]. Based on field investigations, this study selects 54 typical regions of ecological compensation for cultivated land protection as the research objects (Table 1).

### 3.2. Data Sources and Processing

Precise and applied data resources are the foundation of the comprehensive benefit of the external environment of ecological compensation for cultivated land protection. In this study, the descriptive statistical data were taken from the *Chengdu Statistical Yearbook 2017* [46]. The ArcGIS software and Microsoft Excel Software were used to deal with the data. The steps of data processing can be described as follows. The first step deals with data standardization. To eliminate the dimensional effect and the variations in the numerical value of the variable itself. The second step calculates the values of entropy weighting. Step three calculates the values of the comprehensive benefit of external environment of ecological compensation for cultivated land protection. In step four, the Kriging Interpolation Method was used to spatialize the data.

### 3.3. Methods

Variables differ in terms of the units and the degree of variation. To eliminate the dimensional effect and the variations in the numerical value of the variable itself, the data need to be standardized. According to the characteristics of the data, the minimum–maximum normalization method was used to perform this process.
(4)$$Z_{ij}=\frac{X_{ij}-\min X_{j}}{\max X_{j}-\min X_{j}}$$

In Equation (4) [31], *Z_ij_*refers to the standard deviation of the *j-*th index of the *i-*th survey point. *X_ij_*refers to the value of the *j-*th index of the *i-*th survey point. X*_j_* refers to the value of the *j-*th index, max*X_j_* refers to the maximum value of the *j-*th index, and min*X_j_* refers to the minimum value of the *j-*th index.

Based on the standardized data, the information entropy and entropy weight of each index can be defined. Information entropy is related to all possibilities. Every possible event has a probability. Information entropy is the average amount of information we get when an event happens [47]. Thus, mathematically, entropy is the expectation of the amount of information.
(5)$$\begin{array}{l} P_{ij}=Z_{ij}/\sum_{i=1}^{n}Z_{ij}\\ Q_{j}=-\lambda \sum_{i=1}^{n}\left(P_{ij}\ln P_{ij}\right)\\ \lambda =1/\ln n \end{array}$$

In Equation (5) [31], *P_ij_* refers to the ratio of the standard deviation of the *j-*th index to the sum of the standard deviation of this index. *Q_j_* refers to the information entropy of the *j-*th index. *n* =54, Assume that $P_{ij}=0$, $P_{ij}\ln P_{ij}=0$. Using Equation (5), the information entropy of each index can be calculated.

Entropy weight method is used in this study. This method is objective when the index is weighted. According to the variation degree of each index, the entropy weight method calculates the entropy weight of each index by using information entropy [47]. Then, to make the entropy weight more objective, the weight of each index is modified by the entropy weight.
(6)$$\begin{array}{l} U_{j}=(1-Q_{j})/\sum_{j=1}^{m}\left(1-Q_{j}\right)\\ 0\leq U_{j}\leq 1,\\ \sum_{j=1}^{m}U_{j}=1 \end{array}$$

In Equation (6) [31], $U_{j}$ refers to the entropy weight of the *j-*th index, and *m* refers to the number of index factors. On the basis of $Q_{j}$, the entropy weight of the *j-*th index can be calculated. Using to Equation (6), the weights can be calculated in five positive factors and five negative factors (Table 2).

According to the authors’ own illustration drawn from the studied empirical and theoretical literature, the results of the comprehensive level of factors come from the sum of entropy weight multiplied by the standardized data. This means that the comprehensive level of the positive and negative factors in the external environment of ecological compensation for cultivated land protection can be calculated based on the standardized data and entropy weight of each index. Therefore, the comprehensive level of factors can be computed using the following equation:(7)$$F=\sum_{j=1}^{m}U_{j}\cdot Z_{ij}$$

In Equation (7), which is based on the authors’ own illustration drawn from the studied empirical and theoretical literature, *F* refers to the results of the comprehensive level of the positive/negative index factors in the external environmental factors of the cultivated land protection ecological compensation.

## 4. Results

### 4.1. Factor Weights and Comprehensive Benefit Calculation

The factors are divided into positive and negative factor categories. When the value of a positive external factor is larger, it indicates that the factor will increase the revenue for farmers protecting cultivated land. On the contrary, when the value of a negative external factor is larger, it shows that the factor will decrease the revenue for farmers protecting cultivated land. The weights of factors were calculated using Equation (6). In Table 2, the weight of the number of service industries above a designated size in districts (cities) and counties (living environment indicators) is a maximum of 0.219. The weights of general public budget expenditures (living environment indicators) and the number of medical institutions (environmental protection indicators) are 0.142 and 0.128. This shows that farmers pay more attention to issues related to their own lives, employment, and security. This is consistent with the desire of farmers in the new era to pursue a high standard of living, in addition to meeting their material needs. From the weight of the total agricultural output value (production environment index: at least 0.021), it can be seen that farmers do not pay much attention to their own production environment.

### 4.2. Spatialization of Data And Results Analysis

In this study, on the basis of the PSBL, the cost (*TC*) and revenue (*TR*) in the external environment of cultivated land protection ecological compensation can be calculated. The comprehensive benefit of external environment of ecological compensation for cultivated land protection can be calculated by Equation (2). Among the different layers, there are spatial and regional differences in the ecological compensation for cultivated land protection in Chengdu (Table 3).

Based on the ArcGIS10.2 platform, the Kriging Interpolation Method was used to spatialize the data. According to the interpolation results of the selected sample points *B*, the spatial distribution difference of benefit *B* data was obtained.

In Table 3 and Figure 6, the comprehensive benefit from the external environment of ecological compensation for cultivated land protection is highest at urban–rural junction, which is located in the “second layer”. The *B* value is 0.060. However, the *B* value is −0.355 in the central urban area of the “first layer.” The *B* value is −0.049 in the typical rural area of the “third layer”. The result shows that the comprehensive benefit from the external environment of ecological compensation for cultivated land protection has a Λ trend among the different layers. At the same time, *B* > 0 shows that the external environment of the urban–rural junction is favorable for farmers’ willingness to protect cultivated land. To a certain extent, there is a positive effect on farmers’ adoption of cultivated land protection measures. *B* < 0 shows that the external environment situation is relatively poor for cultivated land protection ecological compensation in the central urban area and typical rural area. To a certain extent, the external environment exerts a restraining effect on farmers’ adoption of cultivated land protection measures.

In a typical rural area, the middle-aged and young labor force is mainly made up of migrant workers, while those who stay in the countryside are mainly children and the elderly [48]. Thus, mainly women and the elderly are engaged in agriculture. This situation has been dubbed “women–old agriculture”. There is a large gap between urban and rural areas in terms of human capital [48]. At the time of sowing and harvesting, the cost of migrant workers traveling to and from their hometown to engage in agricultural production is far greater than the amount of government subsidies for cultivated land protection ecological compensation. Thus, some migrant farmers prefer to abandon cultivated land rather than delay work. At the same time, in typical rural areas of China, there is a phenomenon of “getting a grandson and losing a wife.” This refers to when an elderly person goes to the city to take care of their grandchildren, which leads to the loss of some middle-aged and elderly laborers [49]. As a result, there is a phenomenon of labor loss among the middle-aged in typical rural China. As this phenomenon is common in typical rural areas in China, the ecological compensation for cultivated land protection has failed to achieve the expected effect of cultivated land protection.

In the urban–rural junction area, the comprehensive benefit of the external environment for cultivated land protection ecological compensation is positive in the “second layer”. This indicates that, at the urban–rural junction, the external environment from ecological compensation of cultivated land protection has a positive effect on farmers’ willingness to engage in cultivated land protection. On the one hand, as the area is in the transition zone between urban and rural areas, it has advantages over the typical rural area in terms of distance from the central urban area. However, this area is inferior to the central urban area in terms of innovative High-Tech core competitiveness [43,44]. On the other hand, the peasant households in this layer are superior to typical rural peasant households in terms of support for agricultural technology, machinery, and the implementation of new policies. Therefore, there have been many contracted farmers in this layer, and changes in the ecological compensation for cultivated land protection will naturally have a greater impact on the farmers’ willingness to protect the cultivated land in the layer [41].

In the central urban area, as the first layer is far from the countryside, fewer residents were engaged in the primary industry. On the premise that the other conditions remain unchanged, the total agricultural output value is a positive indicator, which means that the residents of the central urban area have a lower agricultural profit. At the same time, as the residents of the central city have lived in the city for a long time, meaning they have less experience of agricultural production [1,2]. Their awareness of the land is weaker than that of typical rural farmers. As a result, the external environment benefit of cultivated land protection ecological compensation is negative in this layer.

## 5. Discussion

Previous research has focused on the quantitative measurement of cultivated land protection externalities [2,3]. In this study, through GIS spatial analysis technology, the regional differences in the comprehensive benefit of urban and rural cultivated land protection ecological compensation were analyzed at the micro scale. Additionally, exploring the influence of different external environments on farmers’ willingness to engage in cultivated land protection in this study. Farmers are the group that are most directly involved in the protection of cultivated land in this study. Their willingness to engage in protection is not only related to individual and family characteristics, but also to their external environment [26,31,36]. To a certain extent, then, the breadth of the research on cultivated land protection ecological compensation has been expanded. In addition, in this study, some factors were considered, such as the geospatial environment, economic development level, employment opportunities. In the urban–rural junction area and the typical rural area, these factors are related to farmers’ willingness to protect cultivated land.

In China, cultivated land serves as both a means of production and as social insurance for farmers. In general, as the key stakeholders of ecological compensation for cultivated land protection, farmers of the typical rural area have a strong enthusiasm to participate in the protection of cultivated land. However, the results of this study indicated that the comprehensive benefit of external environment for cultivated land protection ecological compensation is the highest at the urban–rural junction. In May 2017 and October 2019, this research team did two interviews with farmers by face-to-face. The respondents were not satisfied with the compensation standard of ecological compensation for cultivated land. Under the ecological compensation policy for cultivated land protection, farmers experience a decrease or loss for increasing agricultural input. These losses need to be adequately compensated to sustain their alternative livelihoods and lifestyles [34]. However, local government determines the rate of compensation which was regarded as insufficient by the majority of farmers in previous study [50].

Several policy implications can be drawn from the findings of this study. As such the local governments modify the compensation plan by increasing the compensation standard and improving the compensation mode [51]. The important role of farmer groups in protecting cultivated land, as evidenced in this study, leads to a call for continuous and increased support from the government for farmer group formation when implementing ecological compensation for cultivated land protection. In particular, the government should increase the standard of ecological compensation for cultivated land protection. This will stimulate farmers’ enthusiasm for cultivated land protection. The increased compensation would not only improve farmers’ policy satisfaction, but also reduce resistance to policy implementation. In addition to direct monetary compensation, a long-term supporting scheme also needs to be provided by local authorities to improve farmers in their living level [52]. This supporting scheme could include a better social welfare system, job skills training, and local job opportunities for families in protecting cultivated land. More importantly, a market compensation mechanism could be introduced for ecological compensation for cultivated land protection to guarantee the compensation fund. Furthermore, in order to achieve precise compensation, when the government adopts ecological compensation measures for cultivated land protection, it should pay attention to the external environment in which farmers are located, and analyze the impact of regional differences on farmers’ enthusiasm for cultivated land protection.

In addition, in terms of index selection, there may be some limitations in the selection of external environmental impact factors due to the means of data acquisition and the ease of acquisition. For example, for the selection of the total retail factor of social consumer goods, according to the analysis of relevant economic principles, farmers count as “Limited rational economic people” who pursue the maximization of their own benefits [28,29,31,42]. The higher the total household expenditure on social consumer goods, the less the relative reduction of necessary investment in cultivated land protection (for example, fertilizers, urea, herbicides, etc.), especially for farmers who pay less attention to cultivated land protection. This will indirectly affect the yield of cultivated land [41,53]. As the application of fertilizers, urea, and herbicides is a pre-harvest investment (that is, a pre-investment), the amount of money invested in these areas will affect the expectations of farmers. The psychological incentive can encourage farmers to carry out cultivated land protection. Grain price is generally based on the market supply and demand situation after grain harvesting, which is a kind of after-the-fact adjustment. However, due to the impact of market supply [54] and demand prices [55], if food prices are too low, it will hurt farmers’ crop cultivation in the next season. It is more reasonable to choose the total retail sales of social consumer goods as the representative factor to reflect the market environment. Furthermore, the selection of indicators is based on a comprehensive consideration of external environmental factors. Of course, the index system is still imperfect for cost and income calculation, meaning should be an area for further research in the future.

## 6. Conclusions

As the key stakeholders of ecological compensation for cultivated land protection, farmers’ attitudes, and especially their willingness, are key variables that help determine the effectiveness of polices that seek to coordinate urban–rural development and food security. In this study, GIS spatial analysis technology was used to analyze regional differences and assess ecological compensation for urban and rural cultivated land protection at the micro scale. The results show that the willingness of farmers to participate in cultivated land protection is affected by the external environment of cultivated land protection ecological compensation. The comprehensive benefit from the external environment for cultivated land protection ecological compensation is the highest at the urban–rural junction, which means the comprehensive benefit value is the highest in the “second layer” of Chengdu. The comprehensive benefit shows a Λ trend among the different layers of cultivated land protection ecological compensation in Chengdu. Moreover, in the urban–rural junction area, the external environment is favorable for cultivated land protection ecological compensation in this area. The external environment can increase farmers’ willingness to protect cultivated land and has a positive effect on farmers’ adoption of cultivated land protection measures. In contrast, in the first and third layer, it has a depressant effect on farmers’ willingness to protect cultivated land.

Farmers’ willingness has a significant impact on the implementation of ecological compensation for cultivated land protection, so the government should adopt some incentive measures to improve farmers’ willingness in future implementation of the ecological compensation for cultivated land protection policy.

## Figures and Tables

**Figure 1 ijerph-17-08242-f001:**
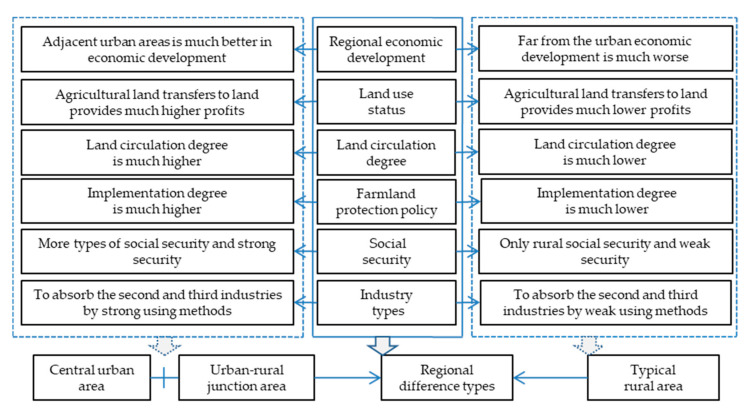
Regional differences of the external environment in terms of ecological compensation for cultivated land protection and.

**Figure 2 ijerph-17-08242-f002:**
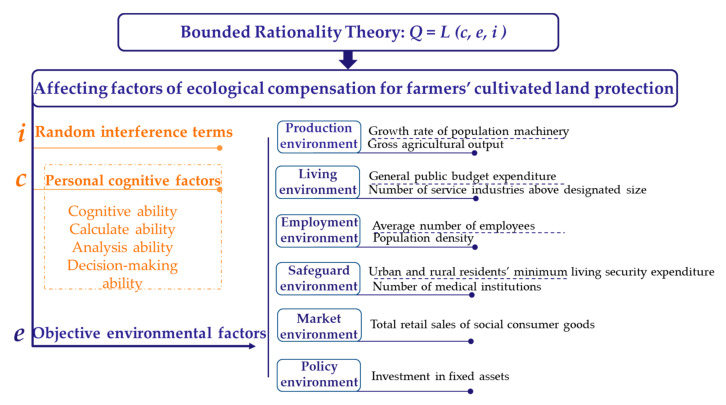
Factors affecting ecological compensation for farmers’ cultivated land protection.

**Figure 3 ijerph-17-08242-f003:**
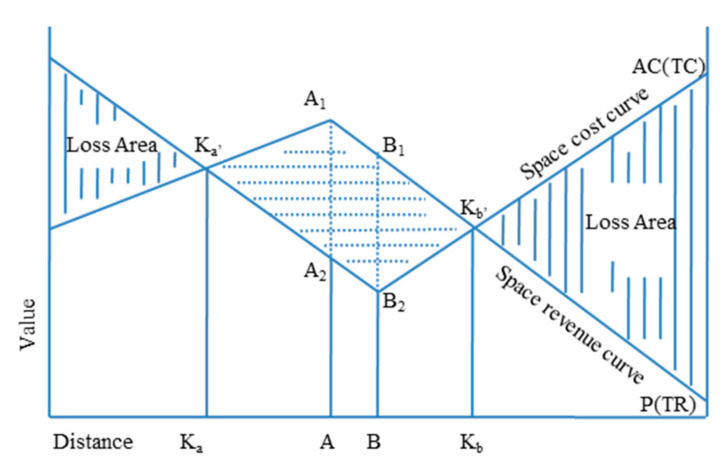
Profitable Spatial Boundary Location model (PSBL).

**Figure 4 ijerph-17-08242-f004:**
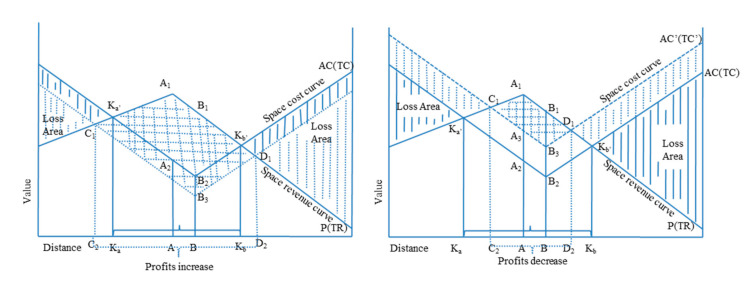
Changes in the model of the spatial boundary between cultivated land protection costs and revenue.

**Figure 5 ijerph-17-08242-f005:**
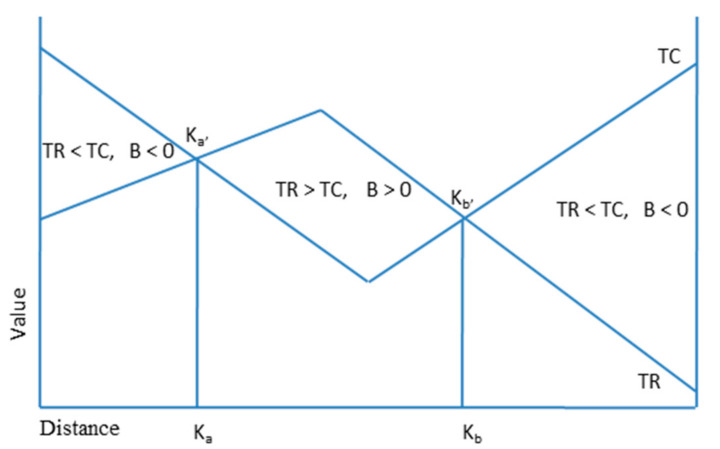
PSBL of external environment of ecological compensation for cultivated land protection.

**Figure 6 ijerph-17-08242-f006:**
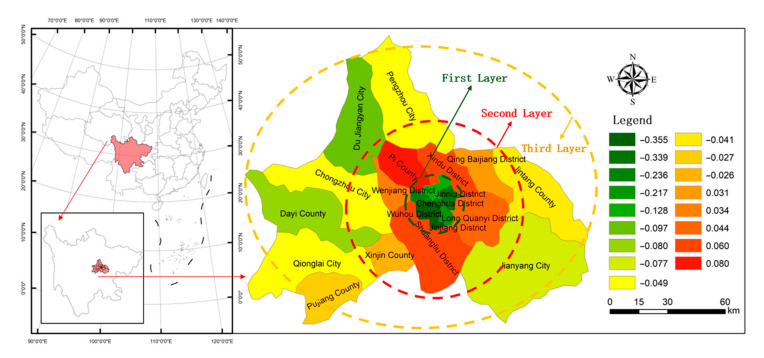
Spatial differences in the comprehensive benefit from the external environment of ecological compensation for cultivated land protection in Chengdu.

**Table 1 ijerph-17-08242-t001:** Typical regions selected within the circle layers for empirical research.

Layer	Prefecture	Jurisdiction	Number of Samples	Distance from Central City	Investigation Area Location Conditions
First Layer	Wuhou District	13 streets	5	Central urban area	High-tech cultural industries, with good location and early “leading development” trend. The sample point is mainly the villages in the city.
Second Layer	Xindu District	13 towns and streets	8	Urban–rural junction area	Industrial agglomeration, good development foundation, a trend of “median development”. The samples are mainly natural villages formed by the adjacent roads.
	Wenjiang District	6 towns and 4 street offices	7		
	Shuangliu District	12 streets(without a hosting area)	8		
Third Layer	Pengzhou City	20 towns and streets	10	Typical rural area	It is an agricultural (mainly basic cultivated land) area, with a poor location, a weak development foundation, and a “successive development
	Qionglai City	8 towns and 6 streets	2		
	Chongzhou City	18 towns and 1 street	14		

**Table 2 ijerph-17-08242-t002:** Factor weights of external environment of ecological compensation for cultivated land protection.

Number	Factors	Attributes	Weights
1	Gross agricultural output (×10^2^ million Yuan)	Revenue/positive factor	0.021
2	Growth rate of population machinery (%)	Revenue/positive factor	0.052
3	Number of medical institutions	Revenue/positive factor	0.128
4	Urban and rural residents’ minimum living security expenditure (×10^4^ Yuan)	Revenue/positive factor	0.048
5	Investment in fixed assets (×10^2^ million Yuan)	Revenue/positive factor	0.122
6	Number of service industries above a designated size	Cost/negative factor	0.219
7	General public budget expenditure (×10^2^ million Yuan)	Cost/negative factor	0.142
8	Population density (person per sq. km.)	Cost/negative factor	0.111
9	Average number of employees (×10^4^ persons)	Cost/negative factor	0.041
10	Total retail sales of social consumer goods (×10^2^ million Yuan)	Cost/negative factor	0.116

Note: data from the Chengdu Statistical Yearbook 2017 [46].

**Table 3 ijerph-17-08242-t003:** Comprehensive benefit from external environment for ecological compensation of cultivated land protection in different layers in Chengdu.

Layers	Sample Location Description	Cost Factors (*TC*)	Revenue Factors (*TR*)	Comprehensive External Environment Benefit (B)
First layer	Central urban area	0.589	0.234	–0.355
Second layer	Urban–rural junction area	0.532	0.592	0.060
Third layer	Typical rural area	0.260	0.211	–0.049

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
