# Peer review of "Regional Differences in Ecological Compensation for Cultivated Land Protection: An Analysis of Chengdu, Sichuan Province, China"

_ijerph, 2020, doi:10.3390/ijerph17218242_

Round 1

Reviewer 1 Report

Review of “Empirical Analysis on Regional Differences of the External Environmental on Cultivated Land Protection Ecological Compensation: Based on the Analysis of 54 Sites in Chengdu, Sichuan Province

Summary:
Authors present a study based on the Profitable Spatial Boundary Analysis Theory (PSBA); they used this theoretical framework to construct the Profitable Spatial Boundary Location Model (PSBL) of the external environment of cultivated land protection ecological compensation. Moreover, they discuss a case study concerning 54 sites in Chengdu, Sichuan Province.

I think that the paper may be accepted with minor revisions. In particular the following corrections are suggested:

  • Line 89: correct “all alternative solution” with “all alternative solutions”
  • Line 91: after the expression “Q = F(c,e,i)” put “where”
  • Figure 1: I suggest explaining the scheme in order to help the reader to immediately understand the information inside the same scheme.
  • Line 129: correct “In the terms of” with “In terms of”
  • Line 157: after introducing the acronym PSBA, it can be used all over the paper, in place of the entire denomination.
  • Line 174: the labels K’a and K’b are indicated as Ka’ and Kb’ in Figure 3, please use the same notation in the text
  • Figure 4: this figure is too small, the labels are unreadable
  • Line 195: is the mention of positive factors correct? Maybe, in this point of the section 3.2. you referred to negative factors
  • Line 268: in this equation you show how to compute the j-th index Qj, hence the summation should be applied to the index i which goes from 1 to n and not to the index j
  • Line 286: correct “Itis" with “It is”.

Author Response

We are grateful to the reviewer for the comments. They are very helpful for guiding us to revise the manuscript. We have carefully revised the manuscript according to the comments. Below we explain in detail how we dealt with the comments that were made.

  • Point 1. Line 89: correct “all alternative solution” with “all alternative solutions”

Response 1: Thank you very much for the comments. As reviewer suggested, we have corrected “all alternative solution” with “all alternative solutions”. (See the Revised Manuscript:Line 86 of Page 2)

  • Point 2. Line 91: after the expression “Q = F (c, e, i)” put “where”

Response 2: Thank you very much for the comments. As reviewer suggested, we put “where” after the expression “Q = F (c, e, i)”. (See the Revised Manuscript:Line 88 of Page 2)

  • Point 3. Figure 1: I suggest explaining the scheme in order to help the reader to immediately understand the information inside the same scheme.

Response 3: Thank you very much for the comments. In order to help the reader to immediately understand the information inside the scheme of Figure 1, we added the following content at the ending of Section 2.1. (See the Revised Manuscript:from Lines 100-114 of Page 3)

Generally speaking, the higher the level of regional economic development in the area, the closer it is to a city. The central urban area and urban‒rural junction area are close to cities and have many industries. However, the closer the farmers are to the town, the more employment opportunities they have, the less dependent they are on land, and the higher the degree of land circulation [33]. On the other hand, the economic development level of the typical rural area is relatively low. In fact, due to farmers’ low level of knowledge and technology, they have fewer opportunities to work in cities and rely more on the land [34]. Therefore, farmers in the typical rural area have a low willingness to transfer land. Moreover, policy is the key factor that prevents cultivated land loss and fragmentation [26]. Some supervision measures are more conducive to the implementation of cultivated land protection policies.

Farmers are the main participants in the implementation of ecological compensation policies for cultivated land protection [35]. The external environment of farmers’ cultivated land protection is different from their own internal factors [36]. The external environment of farmland protection for farmers is affected by market fluctuations and policies. Therefore, this external influence can be improved through measures taken by the government.

Correspondingly, we added and changed the number of references in this manuscript.

33、Zhang H, Zhang Y, Wu S, et al. The Effect of Labor Migration on Farmers' Cultivated Land Quality Protection[J]. Sustainability, 2020, 12.

34、Cheng L, Liu Y, Brown G, et al. Factors affecting farmers' satisfaction with contemporary China's land allocation policy – The Link Policy: Based on the empirical research of Ezhou[J]. Habitat International, 2018: 38-49.

35、Home R, Balmer O, Jahrl I, et al. Motivations for implementation of ecological compensation areas on Swiss lowland farms[J]. Journal of Rural Studies, 2014, 34: 26-36.

36、Sonter L J, Simmonds J S, Watson J E M, et al. Local conditions and policy design determine whether ecological compensation can achieve No Net Loss goals[J]. Nature Communications, 2020, 11(1).

  • Point 4. Line 129: correct “In the terms of” with “In terms of”

Response 4: Thank you very much for the comments. As reviewer suggested, we have corrected “In the terms of” with “In terms of”. (See the Revised Manuscript:Line 137 of Page 4)

  • Point 5. Line 157: after introducing the acronym PSBA, it can be used all over the paper, in place of the entire denomination.

Response 5: Thank you very much for the comments. As reviewer suggested, we have corrected “After introducing the acronym PSBA, it has been used all over the paper” in Line 175 of Page 5, Line 179 of Page 2 and Line 374 of Page 11. (See the Revised Manuscript)

  • Point 6. Line 174: the labels K’a and K’b are indicated as Ka’ and Kb’ in Figure 3, please use the same notation in the text

Response 6: Thank you very much for the comments. As reviewer suggested, we have corrected the labels K’a and K’b” with “Ka’ and Kb’. (See the Revised Manuscript: from Lines 180-182 of Page 5)

  • Point 7. Figure 4: this figure is too small, the labels are unreadable

Response 7: Thank you very much for the comments. As reviewer suggested, in order to make Figure 4 clear, we have enlarged the picture and placed it separately. (See the Revised Manuscript: from Lines 190-191 of Page 6)

  • Point 8. Line 195: is the mention of positive factors correct? Maybe, in this point of the section 3.2. you referred to negative factors

Response 8: Thank you very much for the comments. As reviewer suggested, in order to make more explicit, we have corrected the statement about the factor category. (See the Revised Manuscript: from Lines 195-209 of Page 6)

Because based on the specifical statement of factors in Section “2.2 Factors affecting ecological compensation for farmers' cultivated land protection” and Section “3.1. Theoretical basis”, these factors are divided into positive and negative factor categories. Positive external factors include the factors of the production environment, safeguarding environment, and policy environment. And Negative external factors of cultivated land protection ecological compensation include the living environment, employment environment, and market environment.”

  • Point 9. Line 268: in this equation you show how to compute the j-th index Qj, hence the summation should be applied to the index i which goes from 1 to n and not to the index j

Response 9: Thank you very much for the comments. We have corrected “the index j in the summation” with “the index i in the summation” in equation (4). (See the Revised Manuscript: in Line 282 of Page 9)

                                                                                                             (4)

  • Point 10. Line 286: correct “Itis" with “It is”.

Response 10: Thank you very much for the comments. We have corrected “Itis” with “It is”. (See the Revised Manuscript: Line 306 of Page 10)

We again thank the reviewer for very helpful suggestions throughout the manuscript.

Reviewer 2 Report

Highlight changes in yellow in a next revision, please. No track changes.

Consider comments in the entire text.

I would suggest revising the title and remove “54”

There is no mention to the country…: “Chengdu, Sichuan Province

The beginning is enough to dictate English... and… language revision to the entire text…

Abstract: As an external constraint, the environment of cultivated land protection ecological 12 compensation has important impacts on farmers' willingness and behavior to protect cultivated 13 land.”

A study does not… construct: “this study constructed”

An article does not… make…: “this article makes a systematic analysis”

I do not understand the language used in the text. It needs complete revision--- “score, even the F>0.”

Clarify unclear terms, better than what, why…: “is better,”

Abstract:

Revise it considering brief contextualization and methodology, main findings and practical implications

Revise keywords, remove “city”, add country then…: “Chengdu city”

This is an international journal, authors could avoid reference to explicit politics, there are other ways to mention, and I gave seen many Chinese papers until now:

“The report of the 18th national congress of the Communist Party of China 34 (CPC)”

“Serious differences” mean nothing.,. “There are serious differences between urban and rural areas in China”

Etc:

“is a serious obstacle”

Not a single reference:

2.1. Overall analysis of the external environment

There are serious differences between urban and rural areas in China. The differences are mainly 71 reflected in policies related to household registration and the extent of resource allocation, which 72 is a serious obstacle in China's economic and social development. During the implementation of 73 cultivated land protection ecological compensation policy,farmers are the main participants, whose 74 willingness and actions will affect the effectiveness of policy implementation. However, for 75 cultivated land protection indicates that farmers will lose some potential opportunities, especially 76 farmers in the Urban-Rural Junction Area (URIA). Farmers' cultivated land protection behavior and 77 willingness are not only affected by their own factors, but also by the external environment. These 78 factors mainly include the degree of urban economic development, income level, employment 79 environment, social security, medical benefits, and so on. 80”

And again…

“According to the "Economic Man Hypothesis" theory, human activities are pursuing the 81 maximization of their own interests or utility. However, due to huge population, limited cultivated 82 land resources and strict cultivated land protection policies in China, it is impossible for farmers to 83 fully pursue the goal of maximizing economic benefits. However, farmers are rational. In general, in 84 order to achieve the expected benefits, farmers will make a comprehensive judgment on the external 85 environment of the ecological compensation of cultivated land protection. Therefore, the farmer's 86 cultivated land protection belongs to "Bounded Rationality". 87”

This is not the reference style used before…

“Simon (2001)”

Is this theory or results then?

“From what has been discussed above, there are some regional differences of external 104 environment in cultivated land protection ecological compensation (Figure 1).”

References? “Figure 1. Regional differences of external environment in cultivated land protection and 107 compensation.”

And again, same language: “2.2. Analysis of influencing factors 109

From what has been discussed above, there are some external environment factors are included 110 in the cultivated land protection ecological compensation of farmers (Figure 2).”

And same question, if this is theory, indicate the sources, if already published like that remove…

Figure 2. Affecting factors of ecological compensation for farmers' cultivated land protection.”

Please, NO lists, it is not the international style, easy to write, difficult to connect:

“compensation includes six aspects: Production environment, Living environment, Employment 118 environment, Guarantee environment, Market environment and Policy environment (Figure 2). 119

  • Production Environment”

NO references at all…

“Farmers are the ultimate implementers of the policy. The effect of the cultivated land protection 114 policy is mainly affected by the willingness and behavior of farmers, which are influenced by external 115 environmental factors. According to the unique characteristics of cultivated land, and related 116 research at home and abroad, the external environment of cultivated land protection ecological 117 Int. J. Environ. Res. Public Health 2020, 17, x FOR PEER REVIEW 4 of 13

compensation includes six aspects: Production environment, Living environment, Employment 118 environment, Guarantee environment, Market environment and Policy environment (Figure 2). 119

(1) Production Environment. The main function of cultivated land is to provide food production, 120 so the total agricultural output value is a comprehensive reflection of agricultural production. The 121 higher the regional total agricultural output value is, the better the regional agricultural production 122 condition is. In addition, the stable development of agriculture requires sufficient agricultural labor. 123 Therefore, total agricultural output value and population machinery growth rate are selected to 124 reflect the production environment of ecological compensation for cultivated land protection. 125

(2) Living environment. The greater the number of service industries districts (cities) and 126 counties have, the more attractive they are for rural households to migrate. In fact, a large number of 127 rural laborers have moved to cities and towns, which has indirectly reduced the implementation 128 group of cultivated land protection. In the terms of budget expenditure, as it lacks thorough 129 evaluation mechanism, so it is difficult to match public budget expenditure with social income, or 130 even the net benefits may be negative. However, blindly expanding the scale of budget expenditure 131 will reduce social productivity. Therefore, service industries above designated size and general 132 public budget expenditure are selected to reflect the living environment of the study area. The 133 number of service industries and population machinery growth rate are selected to reflect the 134 production environment of cultivated land protection ecological compensation. 135

(3) Employment environment. In general, population density and the average number of 136 employees are positively correlated with regional employment pressure. Once degree of competition 137 increases, it will cause greater psychological pressure on farmers with weak competitiveness. 138 Therefore, in order to maintain the current living standard, farmers are unwilling to increase the 139 expenditure on cultivated land protection too much. 140

(4) Safeguard environment. Studies have shown that the higher the households' income level is, 141 the more likely farmers are to invest in arable land protection, and the better the social security 142 condition is in rural areas, the less likely they are to give up arable land [31]. Therefore, two indicators 143 are selected to reflect the living environment, which are the number of rural medical and health 144 institutions and the minimum living security expenditure of urban and rural residents. 145

(5) Market environment. When the household income level remains unchanged, the higher the 146 retail sales of consumer goods are, the lower the rural households' investment is in agriculture. At 147 the same time, agricultural production will reduce farmers' enthusiasm for grain production. 148

(6) Policy environment. The increase of the capital stock can promote economic development, 149 adjust the industrial structure, and optimize the environment for implementing the policy of 150 cultivated land protection ecological compensation. Therefore, fixed asset investment is selected as a 151 measure of the policy environment.”

(…)

Not possible

Refeerence?!

3. Theory and Model 153

As "Bounded rationality economic person", based on a comprehensive judgment of the external 154 space environment, farmers will evaluate the costs and benefits of their cultivated land protection 155 behaviors. For evaluation of this situation, the most suitable theory is the Profitable Spatial Boundary 156 Analysis Theory (PSBA), which is proposed by D.M. Smith.”

No reference: “3.1. Theoretical basis

If this is theory and no references, it is considered as absolute original.

Ithenticate will detect all

“” Revise headings: “3.2. Factors attribute definition

Then mathematical data follows, no reference at all, so authors did discover it all alone.. not based in any known work…

It is not the case in 99.99999% of the times…

No formulas but equations ad all know citations must be presented immediately before equations and then parameters and units presented one by one…, as usual, everywhere…

All axis must have legends…

Revise international unit system… “14335km2,” and proper spacing

I advise the authors to revise the language used to be inclusive: “Table 1. Status of 54 survey sites in Chengdu.”

All “sources” must use the reference style, and are indicated in the final list, not like this:

4.2. Data sources 251

According to the previous definition of profit and cost factors, Gross agricultural output, Growth 252 rate of population machinery, Number of medical institutions, Urban and rural residents' minimum 253 living security expenditure, Investment in fixed assets are recorded as P1, P2, P3, P4, P5 respectively. 254 Number of service industries above designated size, General public budget expenditure, Population 255 density, Average number of employees and Total retail sales of consumer goods are recorded as C1, 256 C2, C3, C4, C5 respectively. The data is from Chengdu Statistical Yearbook 2017 (Table 2, 257 http://www.cdstats.chengdu.gov.cn/htm/detail_110939.html).”

Remove “the” and revise the captions: “Table 2. The external environment influencing factors of ecological compensation for protecting cultivated land. 280”

Add all parameters below in notes

Check all other cases

Again, do not use another “listing” style:

“The main reasons are as follows: 317

Firstly”

Etc

Revise language, therefore, therefore…

“Therefore, their awareness of land is 346 weaker than that of typical rural farmers. Therefore, the farmer's external environment benefit of 347 cultivated land protection ecological compensation is negative in this layer.”

If it is a unique section, it cannot be separated, otherwise separate in different sections:

5. Discussion and Conclusion 349

5.1. Discussion

Conclusion: it should contain the same structure addressed to the abstract

See that the manuscript needs extensive further work:

“The combined use of Profitable Spatial Boundary Analysis Theory (PSBA) and the Profitable 381 Spatial Boundary Location Model (PSBL) constitutes a comprehensive framework.”

For what?

“Health” is not addressed at all… but here: “health 144 institutions and the minimum living security expenditure of urban and rural residents”

References: too much focused in the Chinese perspective

2020 references: none, they must be updated…

Just as example

There is a number of significant changes able to be made to the text that will exponentially increase its relevance, Use the comments to do it, compare to international (other than Chinese…) relevant published papers

Reading the title, abstract and conclusions, I would not be convinced t read the entire text, despite the obvious importance of this theme

It is not clear, the importance, innovation and originality of this research. Authors need to highlight it…

Author Response

We are grateful to the reviewer for the comments. They are very helpful for guiding us to revise the manuscript. We have carefully revised the manuscript according to the comments. Below we explain in detail how we dealt with the comments that were made.

  • Point 1

Title: I would suggest revising the title and remove “54”. There is no mention to the country…: “Chengdu, Sichuan Province”

Response 1:

--Thank you very much for the comments.

--As suggested by the reviewer, we have corrected the title “Empirical Analysis on Regional Differences of the External Environmental on Cultivated Land Protection Ecological Compensation: Based on the Analysis of 54 Sites in Chengdu, Sichuan Province” with “Regional Differences in Ecological Compensation for Cultivated Land ProtectionAn Analysis of Chengdu, Sichuan Province, China”. (See the Revised Manuscript: from Lines 2-4 of Page 1)

  • Point 2

Abstract: “Abstract: As an external constraint, the environment of cultivated land protection ecological 12 compensation has important impacts on farmers' willingness and behavior to protect cultivated 13 land.”

(1) A study does not… construct: “this study constructed”

Response 2 - (1):

--Thank you very much for the comments.

--As suggested by the reviewer, we corrected “this study constructed” with “the study adopted”. (See the Revised Manuscript: in Line 13 of Page 1)

(2) An article does not… make…: “this article makes a systematic analysis”

Response 2 - (2):

--Thank you very much for the comments.

--As suggested by the reviewer, we revised “this article makes a systematic analysis” with “The study adopted the Profitable Spatial Boundary Analysis theory (PSBA), using GIS to analyze regional space differences and assess ecological compensation for urban and rural cultivated land protection at the micro scale.” (See the Revised Manuscript: from Lines 13-15 of Page 1)

(3) I do not understand the language used in the text. It needs complete revision---

Clarify unclear terms, better than what, why…: “is better,”

Response 2 - (3):

--We thank the reviewer for pointing out these deficiencies.

--As suggested by the reviewer, in order to make it more explicit in research purpose, innovation, methods and results, we have carefully revised language and logic of the abstract from Lines 11-24 of Page 1. Especially, we have revised and explained “It shows that the external environment of cultivated land protection ecological compensation in URJA is better” in abstract from Lines 17-11 of Page 1.

The trend of the comprehensive benefit of cultivated land protection ecological compensation (B) is "Λ" from the first layer to the third layer. The B value of the urban-rural junction area is the highest value. This shows that the external environment is favorable to ecological compensation in this area, which has a positive effect on farmers’ willingness to protect cultivated land. B<0 in the first and third layer, which has a depressant effect on farmers’ willingness to protect cultivated land.” (See the Revised Manuscript: from Lines 17-22 of Page 1)

The revised abstract is listed as follows:

Abstract: The purpose of this study was to analyze regional differences in ecological compensation for cultivated land protection, and to explore the influence of different external environments on farmers’ willingness to engage in cultivated land protection. The study adopted the Profitable Spatial Boundary Analysis theory (PSBA), using GIS to analyze regional space differences and assess ecological compensation for urban and rural cultivated land protection at the micro scale. The results show that the willingness of farmers to participate in cultivated land protection is affected by the external environment and the ecological compensation offered. The trend of the comprehensive benefit of cultivated land protection ecological compensation (B) is "Λ" from the first layer to the third layer. The B value of the urban-rural junction area is the highest value. This shows that the external environment is favorable to ecological compensation in this area, which has a positive effect on farmers’ willingness to protect cultivated land. B<0 in the first and third layer, which has a depressant effect on farmers’ willingness to protect cultivated land. The study results contribute to our understanding of the impact of regional differences in the external environmental on ecological compensation and farmers’ willingness to engage in cultivated land protection.”

  • Point 3

Keywords: “Revise keywords, remove “city”, add country then…: “Chengdu city”

Response 3:

--We thank the reviewer for pointing out these deficiencies in the keywords section.

--As suggested by the reviewer, we have corrected “Chengdu city” with “China”. The revised abstract is as follows:

Keywords: cultivated land protection; ecological compensation; Profitable Spatial Boundary Model (PSBM); external environmental differences; China (See the Revised Manuscript: from Lines 25-26 of Page 1)

  • Point 4

Introduction: Authors could avoid reference to explicit politics, there are other ways to mention, and I gave seen many Chinese papers until now: “The report of the 18th national congress of the Communist Party of China 34 (CPC)”

Response 4:

--We thank the reviewer for pointing out these deficiencies.

--As suggested by the reviewer, to avoid reference to explicit politics, we have revised all sentences about the explicit politics in this manuscript. Besides, we adopted additional references to demonstrate the political meanings. Correcting the “The report of the 18th national congress of the Communist Party of China 34 (CPC), establishing an ecological compensation mechanism for cultivated land protection is an important measure to achieve coordinated development of social economy and environment” with “Establishing an ecological compensation mechanism for cultivated land protection is an important measure to achieve coordinated development of the social economy and the environment [2,3].” (See the Revised Manuscript: from Lines 33-35 of Page 1)

Correspondingly, we added and changed the number of references in this manuscript.

  • Point 5
  • 1. Overall analysis of the external environment:

(1) “Serious differences” mean nothing.,. “There are serious differences between urban and rural areas in China”

Etc:

“is a serious obstacle”

Not a single reference:

Response 5 - (1):

--Thank you very much for the comments.

--As suggested by the reviewer, we have corrected the language and framework for “2.1. Overall analysis of the external environment”. Specially, correcting “There are serious differences between urban and rural areas in China” with “There are stark differences between urban and rural areas in China [1‒3,27].” (See the Revised Manuscript: in Line 76 Page 2)

In addition, as suggested by the reviewer, we used a more specific statement and added references to demonstrate “a serious obstacle”. Thus, correcting “The differences are mainly reflected in policies related to household registration and the extent of resource allocation, which is a serious obstacle in China's economic and social development.” with “There are stark differences between urban and rural areas in China [1‒3,27]. These differences mainly include the degree of urban economic development, income level, employment environment, social security, medical benefits, and so on [20‒23,25].” (See the Revised Manuscript: from Lines 77-78 Page 2)

Correspondingly, we added and changed the number of references in this manuscript.

(2) This is not the reference style used before…

“Simon (2001)”

Response 5 - (2):

--Thank you very much for the comments.

--As suggested by the reviewer, we have checked and corrected the reference style in this manuscript, correcting the “Simon (2001) pointed out that the rationality of decision-making under "Bounded Rationality" is not absolute optimal solution, but the most satisfactory solution among all alternative solution [29]” with “The rationality of decision-making under “bounded rationality” is not the absolute optimal solution, but is the most satisfactory solution among all alternative solutions [30‒32].” (See the Revised Manuscript: from Lines 85-86 Page 2)

Correspondingly, we added and changed the number of references in this manuscript.

(3) Is this theory or results then?

“From what has been discussed above, there are some regional differences of external 104 environment in cultivated land protection ecological compensation (Figure 1).”

References?

Response 5 - (3):

--We thank the reviewer for pointing out these deficiencies.

--As suggested by the reviewer, this section demonstrates the theorical basis. According to the real differences between urban and rural areas in China, based on the "Economic Man Hypothesis" theory and "Bounded Rationality" theory, we analyzed the external environment of ecological compensation for cultivated land protection from the overall perspective. And to make it more explicit, we have added extensive statement and corresponding references. And the detail theoretical application was added in the ending of “2.1. Overall analysis of the external environment”. (See the Revised Manuscript: from Lines 100-114 Page 3)

The added content is listed as follows:

Generally speaking, the higher the level of regional economic development in the area, the closer it is to a city. The central urban area and urban‒rural junction area are close to cities and have many industries. However, the closer the farmers are to the town, the more employment opportunities they have, the less dependent they are on land, and the higher the degree of land circulation [33]. On the other hand, the economic development level of the typical rural area is relatively low. In fact, due to farmers’ low level of knowledge and technology, they have fewer opportunities to work in cities and rely more on the land [34]. Therefore, farmers in the typical rural area have a low willingness to transfer land. Moreover, policy is the key factor that prevents cultivated land loss and fragmentation [26]. Some supervision measures are more conducive to the implementation of cultivated land protection policies.

Farmers are the main participants in the implementation of ecological compensation policies for cultivated land protection [35]. The external environment of farmers’ cultivated land protection is different from their own internal factors [36]. The external environment of farmland protection for farmers is affected by market fluctuations and policies. Therefore, this external influence can be improved through measures taken by the government.”

In addition, the section of this manuscript has been improved by native speaker.

Terms revision of language:

1) “serious” in Line 76 of Page 2 has been changed to “stark”.

2) “the” in Line 76 of Page 2 has been changed to “These”.

3) “belongs to” in Line 84 of Page 2 has been changed to “adheres to the concept of”.

4) “Bounded Rationality” in Line 84 of Page 2 has been changed to “bounded rationality”.

5) “According to” in Line 92 of Page 2 has been changed to “Based on the”.

6) “Therefore” in Line 93 of Page 2 has been removed.

7) “These differences” in Line 92 of Page 2 has been changed to “These differences may”.

8) In Figure 1, changed the second box from the left to say “Agricultural land transfers to land provides much higher profits”. The one on the right be revised with “lower profits” instead. Capitalized the first word in all boxes. Added “using” after “by strong/weak” in the relevant boxes near the bottom and changed “ability ways” to “methods”.

Correspondingly, we added and changed the number of references in this manuscript.

  • Point 6

2.2. Analysis of influencing factors:

(1) And again, same language: “2.2. Analysis of influencing factors 109

From what has been discussed above, there are some external environment factors are included 110 in the cultivated land protection ecological compensation of farmers (Figure 2).” And same question, if this is theory, indicate the sources, if already published like that remove (Figure 2).

Response 6 - (1):

-- Thank you very much for the comments

--As suggested by the reviewer, this section introduces the factors affecting ecological compensation for farmers’ cultivated land protection, which is a theorical analysis. To make it more clearly, we have added more detailed statement and corresponding references, and revised the Figure 2 as well. (See the Revised Manuscript: from Lines 121-124 of Page 4 and Lines 129-164 of Page 4-5) 

Furthermore, we have corrected “2.2. Analysis of influencing factors” with “2.2. Factors affecting ecological compensation for farmers' cultivated land protection.” (See the Revised Manuscript: in Line 118 of Page 3)

 Correspondingly, we changed the number of references in this manuscript.

(2) Please, NO lists, it is not the international style, easy to write, difficult to connect:

“compensation includes six aspects: Production environment, Living environment, Employment 118 environment, Guarantee environment, Market environment and Policy environment (Figure 2). 119

NO references at all…

Response 6 - (2):

--We thank the reviewer for pointing out these deficiencies.

--As suggested by the reviewer, to demonstrate this question more straightforward regarding the logical framework in this section, we have removed the list style and transformed into international style from Lines 121-164 of Page 4-5.

Based on the "Bounded Rationality" theory, according to the differences in the external environment of the cultivated land protection ecological compensation (Figure 1), we added the "Bounded Rationality" theory application in the Figure 2 to revise it in a more concise way. The Figure 2 is as follows:

Figure 2. Factors affecting ecological compensation for farmers’ cultivated land protection. 

Furthermore, we made a detail statement and added references in this section. (See the Revised Manuscript: from Lines 121-124 of Page 4 and Lines 129-164 of Page 4-5)

Correspondingly, we changed the number of references in this manuscript.

  • Point 7
  1. Theory and Model:

1No reference: “3.1. Theoretical basis”

Revise headings: “3.2. Factors attribute definition”

Response 7 - (1):

-- Thank you very much for the comments

--As suggested by the reviewer, we added references in “3.1. Theoretical basis” section. (See the Revised Manuscript: from Lines 171-189 of Page 5-6)

In addition, we have corrected “3.2. Factors attribute definition” with “3.2. Factors affecting categories of ecological compensation for farmers' cultivated land protection.” Correspondingly, we changed the number of references in this manuscript. (See the Revised Manuscript: in Line 194 of Page 6)

2No formulas but equations ad all know citations must be presented immediately before equations and then parameters and units presented one by one…, as usual, everywhere…

Response 7 - (2):

-- Thank you very much for the comments

--As suggested by the reviewer, to the equation (1) ‒ equation (6) in this manuscript, we carefully checked and modified the format of equations. In particular, in order to distinguish the meaning of F between equation (1) and equation (6), we have corrected “F” with “B” in equation (1), that is, we used “B” refers to the comprehensive benefit of external environment of ecological compensation for cultivated land protection in this study. And we changed and modified all other cases in this manuscript. (See the Revised Manuscript: in Line 215 of Page 6)

Furthermore, we have added references and the meaning of parameters in equations. Correspondingly, we changed the number of references in this manuscript. (See the Revised Manuscript: from Lines 212 -217 of Page 6 and Lines 276-297 of Page 8-9)

  • Point 8
  1. Case study:

(1) Revise international unit system… “14335km2,” and proper spacing

I advise the authors to revise the language used to be inclusive: “Table 1. Status of 54 survey sites in Chengdu.”

Response 8-(1):

-- Thank you very much for the comments

--As suggested by the reviewer, we have corrected “14335km2with “14,335km2, and we have checked and corrected this issue in this manuscript. (See the Revised Manuscript: from Lines 231-232 of Page 7)

Furthermore, we have revised the title of Table 1, which is “Table 1. Status of 54 survey sites in Chengdu.” with “Table 1. Typical regions selected within the circle layers for empirical research”. (See the Revised Manuscript: in Line 257 of Page 8)

Correspondingly, we changed the number of references in this manuscript.

(2) 4.2. Data sources 251

All “sources” must use the reference style, and are indicated in the final list.

Response 8-(2):

-- Thank you very much for the comments

--As suggested by the reviewer, we have removed the unscientific references in this manuscript. All references were indicated in the final list.

In addition, we have corrected “4.2. Data sources” with “4.2. Data sources and process”. At the same time, we revised the statement about data sources. The revised data source are listed as follows (See the Revised Manuscript: from Lines 259-267 of Page 8):

Precise and applied data resources are the foundation of the comprehensive benefit of external environment of ecological compensation for cultivated land protection. In this study, the descriptive statistical data are taken from the Chengdu Statistical Yearbook 2017 (Table 2). The data processing platform uses the ArcGIS software and Microsoft Excel Software, and the basic procedures of data processing are described as follows. The first step deals with data standardization. In order to eliminate the dimensional effect and the variations in the numerical value of the variable itself. The second step calculates the values of entropy weighting. Step three calculates the values of the comprehensive benefit of external environment of ecological compensation for cultivated land protection. Step four the Kriging Interpolation Method was used to spatialize the data.

(3) Remove “the” and revise the captions: “Table 2. The external environment influencing factors of ecological compensation for protecting cultivated land. 280”

Add all parameters below in notes

Response 8-(3):

-- Thank you very much for the comments.

--As suggested by the reviewer, in Line 298 of Page 9, we have corrected “Table 2. The external environment influencing factors of ecological compensation for protecting cultivated land.” With “Table 2. External environment influencing factors of ecological compensation for protecting cultivated land.” In addition, we have improved and added the units of all indictors in Table 2. The revised Table 2 is listed as follows. (See the Revised Manuscript)

Table 2. External environment affecting factors of ecological compensation for protecting cultivated land.

Number

Factors

Attributes

Weights

1

Gross agricultural output(×102 million Yuan)

Revenue / positive factor

0.021

2

Growth rate of population machinery(%)

Revenue / positive factor

0.052

3

Number of medical institutions

Revenue / positive factor

0.128

4

Urban and rural residents' minimum living security expenditure(×104 Yuan)

Revenue / positive factor

0.048

5

Investment in fixed assets(×102 million Yuan)

Revenue / positive factor

0.122

6

Number of service industries above a designated size

Cost / negative factor

0.219

7

General public budget expenditure(×102 million Yuan)

Cost / negative factor

0.142

8

Population density(person per sq. km.)

Cost / negative factor

0.111

9

Average number of employees(×104 persons)

Cost / negative factor

0.041

10

Total retail sales of social consumer goods(×102 million Yuan)

Cost / negative factor

0.116

Note: data from the Chengdu Statistical Yearbook 2017.

(4) Do not use another “listing” style:

“The main reasons are as follows: 317   Firstly” Etc

Revise language, therefore, therefore…

“Therefore, their awareness of land is 346 weaker than that of typical rural farmers. Therefore, the farmer's external environment benefit of 347 cultivated land protection ecological compensation is negative in this layer.”

Response 8-(4):

--We thank the reviewer for pointing out these deficiencies.

--As suggested by the reviewer, in order to make it more explicit in writing style of this section, we have corrected the writing style into a more clearly way. From the spatial perspective of the typical rural area, the urban‒rural junction area and the central urban area, we analyzed the comprehensive benefit of the external environment of ecological compensation for cultivated land protection in different layers. (See the Revised Manuscript: from Lines 337-369 of Page 11)

In addition, the manuscript has been improved by native speaker.

Terms revision of language:

1) “The main reasons are as follows:” in Line 325 of Page 11 has been removed.

2) “Firstly, in typical rural areas” in Line 337 of Page 11 has been changed to “In a typical rural area”.

3) “the current rural middle-aged and young labor force are mainly migrant workers” in Line 338 of Page 11 has been changed to “the middle-aged and young labor force is mainly made up of migrant workers,”.

4) “This situation has been imaged as” in Line 339 of Page 11 has been changed to “This situation has been dubbed”.

5) “Human capital is large gap between urban and rural human capital” in Line 340 of Page 11 has been changed to “There is a large gap between urban and rural areas in terms of human capital”.

6) “At the time of sowing and harvesting, migrant workers, the cost of who traveling to” in Line 341 of Page 11 has been changed to “At the time of sowing and harvesting, the cost of migrant workers traveling to”.

7) “The elderly usually goes to city to take care of grandchildren” in Line 345 of Page 11 has been changed to “This refers to when an elderly person goes to the city to take care of their grandchildren”.

8) “labor loss in middle-aged” in Line 347 of Page 11 has been changed to “labor loss among the middle-aged”.

9) “Secondly, the value of the external environment benefit (F) is positive in the Urban-rural junction of the "Second layer".” from Line 352-352 of Page 11 has been changed to “In the urban‒rural junction area, the comprehensive benefit of the external environment for cultivated land protection ecological compensation is positive in the “second layer”.

10) “farmers' willingness of cultivated land protection” in Line 354 of Page 11 has been changed to “willingness to engage in cultivated land protection”.

11) “Thirdly, as the First layer is far from the countryside,” in Line 363 of Page 11 has been changed to “In the central urban area, as the first layer is far from the countryside,”.

12) “the first industry” in Line 364 of Page 11 has been changed to “the primary industry”.

13) “Therefore, the farmer's external environment benefit of cultivated land protection ecological compensation is negative in this layer.” in Line 368 of Page 11 has been changed to “As the result, the external environment benefit of cultivated land protection ecological compensation is negative in this layer”.

Correspondingly, we changed the number of references in this manuscript.

  • Point 9
  1. Discussion and Conclusion:

Conclusion: it should contain the same structure addressed to the abstract

Response 9:

-- Thank you very much for the comments.

--As suggested by the reviewer, we have carefully improved the “Conclusion”, which making the conclusion contain the same structure addressed to the abstract from Lines 423-443. We have corrected the language and framework for this section. And in the ending of conclusion, we added policy implications. (See the Revised Manuscript: from Lines 444-455 of Page 13)

The conclusion is listed as follows:

“This study analyzed regional differences in external environment and ecological compensation for cultivated land protection to explore the influence of different factors on farmers’ willingness to engage in cultivated land protection, using 54 investigation points data from Chengdu in southwest China. Using GIS, the external environment, influencing factors, and comprehensive benefits of ecological compensation for cultivated land protection were examined and discussed. Furthermore, we examined the external environment of ecological compensation for cultivated land protection to show the circle layer distribution. 

The results show that the willingness of farmers to participate in cultivated land protection is affected by the external environment of cultivated land protection ecological compensation. The comprehensive benefit of external environment for cultivated land protection ecological compensation is the highest at urban-rural junction, which means the comprehensive benefit B value (0.060) is the highest in the “second layer” of Chengdu. However, the B value is ‒0.355 in the central urban area of the “first layer,” and ‒0.049 in the typical rural area of the “third layer.” Therefore, the comprehensive benefit shows a Λ trend among the different layers of cultivated land protection ecological compensation in Chengdu. Moreover, in the urban‒rural junction area, the comprehensive benefit of the external environment for cultivated land protection ecological compensation is B > 0. This shows that the external environment is favorable for cultivated land protection ecological compensation in this area. The external environment can increase farmers’ willingness to protect cultivated land and has a positive effect on farmers’ adoption of cultivated land protection measures. In contrast, in the first and third layer, it has a depressant effect on farmers’ willingness to protect cultivated land.

A number of policy implications can be drawn from the findings of this study. The important role of farmer groups in protecting cultivated land, as evidenced in this study, leads to a call for continuous and increased support from the government for farmer group formation when implementing ecological compensation for cultivated land protection. In particular, the government should increase the standard of ecological compensation for cultivated land protection. This will stimulate farmers’ enthusiasm for cultivated land protection. In addition, in order to achieve precise compensation, when the government adopts ecological compensation measures for cultivated land protection, it should pay attention to the external environment where farmers are located, and analyze the impact of regional differences on farmers’ enthusiasm for cultivated land protection. In addition, farmers’ willingness has a significant impact on the implementation of ecological compensation for cultivated land protection, so the government should adopt some incentive measures to increase farmers’ willingness to protect cultivated land.”

In addition, in terms of language and framework, we have made major changes to discussion. The revised discussion is listed as follows (See the Revised Manuscript: from Lines 372-421 of Page 11-13):

To our knowledge, previous research has focused on the quantitative measurement of cultivated land protection externalities [2-3]. In using GIS and adopting the PSBA, this study performs an analysis of regional differences in the comprehensive benefit of urban and rural cultivated land protection ecological compensation at the micro scale. It also explores the influence of different external environments on farmers’ willingness to engage in cultivated land protection. This study considers that farmers are the group that are most directly involved in the protection of cultivated land. Their willingness to engage in protection is not only related to individual and family characteristics, but also to their external environment [26,31,36]. To a certain extent, then, the breadth of the research on cultivated land protection ecological compensation has been expanded. In addition, we considered factors such as the geospatial environment, economic development level, employment opportunities, and so on. In urban‒rural junction area and typical rural area, these factors are related to farmers’ willingness to protect cultivated land.

In China, cultivated land serves as both a means of production and as social insurance for farmers. In general, as the key stakeholders of ecological compensation for cultivated land protection, farmers of the typical rural area have a strong enthusiasm to participate in the protection of cultivated land. However, our data indicated that the comprehensive benefit of external environment for cultivated land protection ecological compensation is the highest at urban-rural junction. In May 2017 and October 2019, we did two interviews with farmers by face-to-face. The respondents were not satisfied with the compensation standard of ecological compensation for cultivated land. Under the ecological compensation policy for cultivated land protection, farmers experience a decrease or loss for increasing agricultural input. These losses need to be adequately compensated to sustain their alternative livelihoods and lifestyles [34]. However, local government determines the rate of compensation which was regarded as insufficient by the majority of farmers in previous study [48], The result of study is consistent with our conclusion. As such, we propose that local governments modify the compensation plan by increasing the compensation standard and improving the compensation mode [49]. The increased compensation would not only improve farmers' policy satisfaction, but also reduce resistance to policy implementation. In addition to direct monetary compensation, a long-term supporting scheme also needs to be provided by local authorities to improve farmers in their living level [50]. This supporting scheme could include a better social welfare system, job skills training, and local job opportunities for families in protecting cultivated land. More importantly, a market compensation mechanism could be introduced for ecological compensation for cultivated land protection to guarantee the compensation fund.

In terms of index selection, there may be some limitations in the selection of external environmental impact factors due to the means of data acquisition and the ease of acquisition. For example, for the selection of the total retail factor of social consumer goods, according to the analysis of relevant economic principles, farmers count as “Limited rational economic people” who pursue the maximization of their own goals and benefits [28,29,31,42]. The higher the total household expenditure on social consumer goods, the less the relative reduction of necessary investment in cultivated land protection (for example, fertilizers, urea, herbicides, etc.) [41], especially for farmers who pay less attention to cultivated land protection. This will indirectly affect the yield of cultivated land [41,51]. As the application of fertilizers, urea, and herbicides is a pre-harvest investment (that is, a pre-investment), the amount of money invested in these areas will affect the expectations of farmers. The psychological incentive can encourage farmers to carry out cultivated land protection. Grain price is generally based on the market supply and demand situation after grain harvesting, which is a kind of after-the-fact adjustment. However, due to the impact of market supply and demand prices [52,53], if food prices are too low, it will hurt farmers’ crop cultivation in the next season. It is more reasonable to choose the total retail sales of social consumer goods as the representative factor to reflect the market environment. Furthermore, the selection of indicators is based on a comprehensive consideration of external environmental factors. Of course, the index system is still imperfect for cost and income calculation, meaning should be an area for further research in the future.

  • Point 10
  1. Further work:

See that the manuscript needs extensive further work:

(1) “The combined use of Profitable Spatial Boundary Analysis Theory (PSBA) and the Profitable 381 Spatial Boundary Location Model (PSBL) constitutes a comprehensive framework.”

For what?

Response 10-(1):

-- Thank you very much for the comments.

--As suggested by the reviewer, to make it more explicit, we have removed the vague statement in conclusion. We made a further explanation about the issue at the beginning of the conclusion. The revised further explanation is listed as follows (See the Revised Manuscript: from Lines 423-429 of Page 12-13)

“This study analyzed regional differences in external environment and ecological compensation for cultivated land protection to explore the influence of different factors on farmers’ willingness to engage in cultivated land protection, using 54 investigation points data from Chengdu in southwest China. Using GIS, the external environment, influencing factors, and comprehensive benefits of ecological compensation for cultivated land protection were examined and discussed. Furthermore, we examined the external environment of ecological compensation for cultivated land protection to show the circle layer distribution.” as the beginning of the Conclusion.

(2) “Health” is not addressed at all… but here: “health 144 institutions and the minimum living security expenditure of urban and rural residents”

2020 references: none, they must be updated…

Response 10-(2):

-- Thank you very much for the comments.

--As suggested by the reviewer, we have added health-related statement in this manuscript. Specially, from the food security perspective, we demonstrated the importance of food security and human survival, as well as the impact of the destruction of cultivated land on the human living environment.

1) The statement was added in introduction as follows:

In developing countries, there is a significant income gap between the nonagricultural employment and agricultural employment [1] The high income from nonagricultural employment attracts many agricultural laborers to work in the city [1,2]. Thus, the cultivated land is poorly protected, and is abandoned, which results in food security has a threat in developing countries [3]. In addition, the destruction of cultivated land ecosystem will negative affect the ecological environment needed for human survival as well [2,3].” (See the Revised Manuscript: from Lines 28-33 of Page 1)  

2) The statement was added in “2.2. Factors affecting ecological compensation for farmers' cultivated land protection.” as follows:

The improvement of the rural social security mechanism is conducive to an increase in farmers’ willingness to protect cultivated land [31]. In 2008, Chengdu was the pilot area for exploring the cultivated land protection ecological compensation system in China. In order to implement the cultivated land protection ecological compensation policy, Chengdu set up a special fund for farmers who protect cultivated land. This special cultivated land protection fund was for farmers to purchase endowment insurance [25]. The purpose is to provide livelihood security and encourage farmers to protect cultivated land. Farmers’ pension issues are also considered, which is a highlight of the ecological compensation policy in Chengdu [21,25]”. (See the Revised Manuscript: from Lines 149-155 of Page 4-5)  

3) The statement was added in discussion as follows:

In China, cultivated land serves as both a means of production and as social insurance for farmers. In general, as the key stakeholders of ecological compensation for cultivated land protection, farmers of the typical rural area have a strong enthusiasm to participate in the protection of cultivated land. However, our data indicated that the comprehensive benefit of external environment for cultivated land protection ecological compensation is the highest at urban-rural junction. In May 2017 and October 2019, we did two interviews with farmers by face-to-face. The respondents were not satisfied with the compensation standard of ecological compensation for cultivated land. Under the ecological compensation policy for cultivated land protection, farmers experience a decrease or loss for increasing agricultural input. These losses need to be adequately compensated to sustain their alternative livelihoods and lifestyles [34]. However, local government determines the rate of compensation which was regarded as insufficient by the majority of farmers in previous study [48], The result of study is consistent with our conclusion. As such, we propose that local governments modify the compensation plan by increasing the compensation standard and improving the compensation mode [49]. The increased compensation would not only improve farmers' policy satisfaction, but also reduce resistance to policy implementation. In addition to direct monetary compensation, a long-term supporting scheme also needs to be provided by local authorities to improve farmers in their living level [50]. This supporting scheme could include a better social welfare system, job skills training, and local job opportunities for families in protecting cultivated land. More importantly, a market compensation mechanism could be introduced for ecological compensation for cultivated land protection to guarantee the compensation fund.” (See the Revised Manuscript: from Lines 384-403 of Page 12)

In addition, we have added and updated the 2020 references in this manuscript. The 2020 references as follows:

1、El-Osta H S . The rural–urban income divide among farm households: the role of off-farm work and farm size[J]. Agricultural Finance Review, 2020, 80(4):453-470.

4、 Niu H P, Xiao D Y, Wang K P. Quantification of externality of cultivated land protection and its Compensation effect in major grain-producing areas [M]. Beijing: Science Press, 2020.

33、Zhang H, Zhang Y, Wu S, et al. The Effect of Labor Migration on Farmers' Cultivated Land Quality Protection[J]. Sustainability, 2020, 12.

36、Sonter L J, Simmonds J S, Watson J E M, et al. Local conditions and policy design determine whether ecological compensation can achieve No Net Loss goals[J]. Nature Communications, 2020, 11(1).

37、Wang X, Xin L, Tan M, et al. Impact of spatiotemporal change of cultivated land on food-water relations in China during 1990-2015[J]. The Science of the Total Environment, 2020, 716(5): 137119.1-137119.11.

49、Awal A R, Awudu A. Farmer groups, collective marketing and smallholder farm performance in rural Ghana [J]. Journal of Agribusiness in Developing and Emerging Economies, 2020, 10(5): 511-527.

(3) “There is a number of significant changes able to be made to the text that will exponentially increase its relevance, Use the comments to do it, compare to international (other than Chinese…) relevant published papers

It is not clear, the importance, innovation and originality of this research. Authors need to highlight it…

Response 10-(3):

-- Thank you very much for the comments.

--As suggested by the reviewer, we have made a number of significant changes in language, structure and framework. Especially, we have highlighted the innovation of this research in “Introduction” and “Discussion” section. These changes are marked in yellow in this manuscript.

The innovation of this research was highlighted in introduction:

we want to investigate the internal mechanism that affects the willingness of farmers to protect cultivated land in terms of the external environmental differences by regions. This will allow us to understand how external environmental differences affect farmers’ willingness to engage in cultivated land protection. These issues are still not clarified.” (See the Revised Manuscript: from Lines 64-67 of Page 1)

The innovation of this research was highlighted in discussion:

To our knowledge, previous research has focused on the quantitative measurement of cultivated land protection externalities [2-3]. In using GIS and adopting the profitable spatial boundary analysis theory (PSBA), this study performs an analysis of regional differences in the comprehensive benefit of urban and rural cultivated land protection ecological compensation at the micro scale. It also explores the influence of different external environments on farmers’ willingness to engage in cultivated land protection. This study considers that farmers are the group that are most directly involved in the protection of cultivated land. Their willingness to engage in protection is not only related to individual and family characteristics, but also to their external environment [26,31,36]. To a certain extent, then, the breadth of the research on cultivated land protection ecological compensation has been expanded. In addition, we considered factors such as the geospatial environment, economic development level, employment opportunities, and so on. In urban‒rural junction area and typical rural area, these factors are related to farmers’ willingness to protect cultivated land.” (See the Revised Manuscript: from Lines 372-383 of Page 11-12)

We again thank the reviewer for very helpful suggestions throughout the manuscript.

Round 2

Reviewer 2 Report

Highlight changes in yellow in a next revision, please. No track changes.

Consider comments in the entire text.

Abstract should start by contextualizing the study…

 Abstract: The purpose of this study was to analyze regional differences in ecological compensation 11 for cultivated land protection, and to explore the influence of different external environments on 12 farmers’ willingness to engage in cultivated land protection.”

Again [as before…], a study does not adopt… “The study adopted the Profitable”

Revise spacing: “B<0”

Do not use “our” or other personal expressions… “to our” “We”…

When the same references are continuously cited, remove the first, leave the last…

“ [2,3]. Establishing an ecological compensation 33 mechanism for cultivated land protection is an important measure to achieve coordinated 34 development of the social economy and the environment [2,3].”

[And again…

“[43,44]. Smith combined space cost curve theory by Weber and space 173 revenue curve theory by Losch [43,44],”]

Parameters and units are resented after… equations, not before…. And if the equation os already known, then the reference must be cited immediately before introducing the equation to be presented below in the text

References like this need to connect to a reference number… the style used in this journal… “Chengdu Statistical Yearbook 2017 (Table 2).

Again, this is not a correct reference, add the reference number too…

“Note: data from the Chengdu Statistical Yearbook 2017.”

Further comments:

Remove it since this is not a thesis…

“The rest of the paper is organized as follows. Section 2 is an analysis of the external environment and how it affects cultivated land protection ecological compensation. Section 3 describes the 69 theoretical basis and model. The data and descriptive statistics of the variables used in the analysis 70 are presented in Section 4, as well as the empirical specifications. A discussion of the empirical results 71 follows in Section 5. The final section presents the conclusions and policy implications of the study. 72”

Huge heading. Make it more brief…

2. Analysis of the external environment and how it affects cultivated land protection ecological compensation 74”

This is an equation, to be presented outside the sentence… and properly cited…

“Bounded rationality” is defined as (,e, i) Q Lc=”

After this, NO text…! But a Figure… Not possible

2.2. Factors affecting ecological compensation for farmers' cultivated land protection

Confusing location of references, if the Fugue is published, it must be removed…

“(Figure 3) [30‒32].”

Figure 4 relates two different figures, so details the sub captions in the main caption… Same comments as before too, if known, no added knowledge… to me removed…

Figure 4. Changes in the model of the spatial boundary between cultivated land protection costs and 192 revenue.”

““2.2. Factors affecting ecological compensation for farmers' cultivated land protection

And then

3.2. Factors affecting categories of ecological compensation for farmers' cultivated land protection.”?! It seems it would be in the same section then…

Given the similarity of headings…

Confusing structure… See:

3. Theoretical analysis

3.1. Theoretical basis

3.2. Factors affecting categories of ecological compensation for farmers' cultivated land protection.

3.3. Theoretical application

Process of what? or procedure?!

4.2. Data sources and process

Equations must have numbers and citation immediately next to reference number…

“[31], the equation was as follows: 273”

All equations must be “introduced” in the text… by number…

“According to the standardized data, the information entropy of the j-th index can be defined as Qj : 280”

Etc…

(…)

The term “factors” should be further contextualized here and in the text and headings…

Table 2. External environment affecting factors of ecological compensation for protecting cultivated land. 298”

Revise English, possessive case…

5.1. Factors weights and comprehensive benefit calculation

Alerted before, check all similar language

“obvious” means exactly what?!

“are obvious spatial”

When the title is together, there cannot be any separation by headings, otherwise separate in different sections…

6. Discussion and Conclusion

Use plural in Conclusions…

Previously stated…

Conclusions needs brief contextualization and methodology

References

“size[J].”?! and many others… “areas [M].”

The reference style is not the one followed by the journal

Pages? “Beijing: Science Press, 2020.”

The authors made an effort to improve the text. But id still needs additional work.

Author Response

  • Point 1

Abstract: The purpose of this study was to analyze regional differences in ecological compensation 11 for cultivated land protection, and to explore the influence of different external environments on 12 farmers’ willingness to engage in cultivated land protection.”

(1) Abstract should start by contextualizing the study…

Response 1 - (1):

--Thank you very much for the comments.

--As suggested by the reviewer, we have added the contextualizing background at the beginning of the abstract. The added contextualizing background is listed as follows:

Exploring the elements that affect farmers’ willingness to protect cultivated land is the key to improving the ecological compensation mechanism for cultivated land protection.”. (See the Revised Manuscript: from Lines 11-12 of Page 1)

(2) a study does not adopt… “The study adopted the Profitable”

Response 1 - (2):

--Thank you very much for the comments.

--As suggested by the reviewer, we revised “The study adopted the Profitable Spatial Boundary Analysis theory (PSBA), using GIS to analyze regional space differences and assess ecological compensation for urban and rural cultivated land protection at the micro scale.” with “Based on the Profitable Spatial Boundary Analysis theory (PSBA), the study used GIS to analyze regional space differences and assess ecological compensation for urban and rural cultivated land protection at the micro scale.” (See the Revised Manuscript: from Lines 15-16 of Page 1)

(3) Revise spacing: “B<0”

Response 1 - (3):

--We thank the reviewer for pointing out this deficiency.

--As suggested by the reviewer, we have carefully revised “B<0” with “B < 0” in this manuscript in Line 23 of Page 1.

  • Point 2

Do not use “our” or other personal expressions… “to our” “We”…

Response 2:

--We thank the reviewer for pointing out these deficiencies in this manuscript.

--As suggested by the reviewer, we have removed all personal expressions.

Terms revision of language:

1) “our” in Line 25 of Page 1 has been changed to “the”.

2) “we want” in Line 66 of Page 2 has been changed to “this study wants”.

3) “we” has been changed to “this study” in Line 128 of Page 4, Line 311 of Page 9, Line 379 of Page 11, Line 386 of Page 11, and Line 395 of Page 11.

4) “To our knowledge,” in Line 371 of Page 11 has been removed.

  • Point 3

When the same references are continuously cited, remove the first, leave the last…

“ [2,3]. Establishing an ecological compensation 33 mechanism for cultivated land protection is an important measure to achieve coordinated 34 development of the social economy and the environment [2,3].”

[And again…

“[43,44]. Smith combined space cost curve theory by Weber and space 173 revenue curve theory by Losch [43,44],”]

Response 3:

--We thank the reviewer for pointing out these deficiencies.

--As suggested by the reviewer, when the same references are continuously cited, we have removed the first, and left the last.

Correcting the “[2,3]. Establishing an ecological compensation mechanism for cultivated land protection is an important measure to achieve coordinated development of the social economy and the environment [2,3].” with “Establishing an ecological compensation mechanism for cultivated land protection is an important measure to achieve coordinated development of the social economy and the environment [2,3].” (See the Revised Manuscript: from Lines 34-35 of Page 1)

Correcting the “[43,44]. Smith combined space cost curve theory by Weber and space revenue curve theory by Losch [43,44],” with “Smith combined space cost curve theory by Weber and space revenue curve theory by Losch [43,44],” (See the Revised Manuscript: from Lines 170-171 of Page 5)

  • Point 4

(1) Parameters and units are resented after… equations, not before…. And if the equation os already known, then the reference must be cited immediately before introducing the equation to be presented below in the text

Equations must have numbers and citation immediately next to reference number…

“[31], the equation was as follows: 273

All equations must be “introduced” in the text… by number…

“According to the standardized data, the information entropy of the j-th index can be defined as Qj : 280”

Etc…

(…)

Response 4 - (1):

--Thank you very much for the comments.

--As suggested by the reviewer, we have corrected the unscientific writing style in corresponding parameters and units.

We have removed “In Equations (2), B represents the comprehensive benefit of external environment of ecological compensation for cultivated land protection” from Lines 207-208 of Page 6. And we put “B represents the comprehensive benefit of external environment of ecological compensation for cultivated land protection” in “In Equations (2) and (3) [30,43,44], B represents the comprehensive benefit of external environment of ecological compensation for cultivated land protection,” (See the Revised Manuscript: from Lines 210-211 of Page 6.)

In addition, we checked all the equations in this manuscript. (See the Revised Manuscript: from Lines 208-209 of Page 6 and from Lines 267-286 of Page 8)  

Terms revision of language and references:

1) “the information entropy of the j-th index can be defined as Qj :” in Line 272 of Page 8 has been changed to “the information entropy can be defined.”.

2) “the entropy weight Uj of the j-th index can be calculated [31],” in Line 283 of Page 8 has been changed to “the entropy weight of the j-th index can be calculated [31],”. And we put “Uj refers to the entropy weight of the j-th index,” in “In Equation (7), Uj refers to the entropy weight of the j-th index,” (See the Revised Manuscript: from Lines 282-285 Page 8)

3) we have removed the statement “With the goal of standardizing the data [31], the equation was as follows: ” (See the Revised Manuscript: in Line 273-274 of Page 8).

(2) References like this need to connect to a reference number… the style used in this journal… “Chengdu Statistical Yearbook 2017 (Table 2).

Again, this is not a correct reference, add the reference number too…

Response 4 - (2):

--Thank you very much for the comments.

--As suggested by the reviewer, we have checked and corrected the reference style in this manuscript, correcting the “Chengdu Statistical Yearbook 2017 (Table 2)” with “Chengdu Statistical Yearbook (2017) [46]” (See the Revised Manuscript: in Line 255 Page 7)

Correspondingly, we added and changed the number of references in this manuscript: “46. National Bureau of Statistics of China. China Statistical Yearbook; China Statistics Press: Beijing, China, 2017.” (See the Revised Manuscript: in Line 564 Page 15)

In addition, we added the reference number in “Note: data from the Chengdu Statistical Yearbook (2017) [46].” (See the Revised Manuscript: in Line 309 Page 9)

(3) Remove it since this is not a thesis…

“The rest of the paper is organized as follows. Section 2 is an analysis of the external environment and how it affects cultivated land protection ecological compensation. Section 3 describes the 69 theoretical basis and model. The data and descriptive statistics of the variables used in the analysis 70 are presented in Section 4, as well as the empirical specifications. A discussion of the empirical results 71 follows in Section 5. The final section presents the conclusions and policy implications of the study. 72”

Response 4 - (3):

--We thank the reviewer for pointing out the deficiency.

--As suggested by the reviewer, we have removed this section from Lines 68-72 Page 2 in this manuscript.

  • Point 5

(1) Huge heading. Make it more brief …

“2. Analysis of the external environment and how it affects cultivated land protection ecological compensation 74”

(2) Confusing structure… See:

“3. Theoretical analysis”

“3.1. Theoretical basis”

“3.2. Factors affecting categories of ecological compensation for farmers' cultivated land protection.

“3.3. Theoretical application”

(3) “2.2. Factors affecting ecological compensation for farmers' cultivated land protection”

“3.2. Factors affecting categories of ecological compensation for farmers' cultivated land protection.”?! It seems it would be in the same section then…

Given the similarity of headings…

(4) When the title is together, there cannot be any separation by headings, otherwise separate in different sections…

“6. Discussion and Conclusion”

Response 5:

-- Thank you very much for the comments

--As suggested by the reviewer, to make it more clearly and explicit, we have carefully modified the framework of this manuscript. In particular, due to Section 2 and Section 3 are belong to theoretical analysis of external environment of ecological compensation for cultivated land protection, we merged Section 2 and Section 3. Correspondingly, we have revised the number and order of headings in this manuscript.

The revised the framework of this manuscript is listed as follows:

“1. Introduction

  1. Theoretical analysis

2.1. External environment of ecological compensation for cultivated land protection

2.2. Factors selection

2.3. Profitable Spatial Boundary Analysis Theory (PSBA)

2.4. Classification of factors based on PSBA

2.5. Model specification based on PSBA

  1. Materials and Methods

3.1. Study area

3.2. Data sources and processing

3.3. Methods

  1. Results

4.1. Factors’ weights and comprehensive benefit calculation

4.2. Spatialization of data and results analysis

  1. Discussion
  2. Conclusions”

(See the Revised Manuscript)

Response 5-(1):

-- Thank you very much for the comments

--As suggested by the reviewer, according to the revised framework, we have changed “2.2. Factors affecting ecological compensation for farmers' cultivated land protection” to “2.2 Factors selection”. (See the Revised Manuscript: in Line 115 of Page 3)

Response 5-(2):

-- Thank you very much for the comments

--As suggested by the reviewer, according to the revised framework, we have changed “3.1. Theoretical basis” to “2.3. Profitable Spatial Boundary Analysis Theory (PSBA)” (See the Revised Manuscript: in Line 163 of Page 5), changed “3.2. Factors affecting categories of ecological compensation for farmers' cultivated land protection.” to “2.4. Classification of factors based on PSBA” (See the Revised Manuscript: in Line 189 of Page 5), and changed “3.3. Theoretical application” to “2.5. Model specification based on PSBA” (See the Revised Manuscript: in Line 205 of Page 6).

Response 5-(3):

-- Thank you very much for the comments

--As suggested by the reviewer, according to the revised framework, we have changed “3.2. Factors affecting categories of ecological compensation for farmers' cultivated land protection” to “2.4. Classification of factors based on PSBA”. (See the Revised Manuscript: in Line 189 of Page 5)

Response 5-(4):

-- Thank you very much for the comments

--As suggested by the reviewer, we have changed “6. Discussion and Conclusion” to “5. Discussion” (See the Revised Manuscript: in Line 370 of Page 11) and “6. Conclusions” (See the Revised Manuscript: in Line 431 of Page 12)

  • Point 6

This is an equation, to be presented outside the sentence… and properly cited…

“Bounded rationality” is defined as (,e, i) Q Lc=”

Response 6:

--We thank the reviewer for pointing out these deficiencies.

--As suggested by the reviewer, we have put the equation outside the sentence and revised the cited style. Correspondingly, we added and changed the number of equations in this manuscript (See the Revised Manuscript: from Lines 85-86 of Page 2.)

The revised equation is listed as follows:

Bounded rationality” is defined as fallow:

                                          (1)

In equation (1), c represents the cognitive factors of the decision-making subject, e represents the external environment factors, and i represents the random interference terms [30,31].

  • Point 7

(1) After this, NO text…! But a Figure… Not possible

“2.2. Factors affecting ecological compensation for farmers' cultivated land protection”

Response 7 - (1):

-- Thank you very much for the comments

--As suggested by the reviewer, due to all the text in “2.2 Factors selection” is an explanation of the contents of Figure 2, to make it more clearly, we have put the Figure 2 in the end of “2.2 Factors selection”.

2Confusing location of references, if the Fugue is published, it must be removed…

“(Figure 3) [30‒32].”

Response 7 - (2):

-- Thank you very much for the comments

--As suggested by the reviewer, we carefully checked and modified the location of references. In particular, we have changed “through space boundary analysis to find the best location (Figure 3) [30‒32]” to “through space boundary analysis to find the best location [30‒32]”. (See the Revised Manuscript: in Line 171 of Page 5)

  • Point 8

Figure 4 relates two different figures, so details the sub captions in the main caption… Same comments as before too, if known, no added knowledge… to me removed…

“Figure 4. Changes in the model of the spatial boundary between cultivated land protection costs and 192 revenue.”

Response 8:

-- Thank you very much for the comments

--As suggested by the reviewer, we have modified Figure 4 to make it the same as the first submitted manuscript. (See the Revised Manuscript: in Line 186 of Page 5)

Figure 4. Changes in the model of the spatial boundary between cultivated land protection costs and revenue.

  • Point 9

Process of what? or procedure?!

“4.2. Data sources and process”

Response 9:

-- Thank you very much for the comments.

--As suggested by the reviewer, to make more it clearly, we have changed the “Data sources and process” to “Data sources and processing”. (See the Revised Manuscript: in Line 252 of Page 7)

In this study, the data are taken from the Chengdu Statistical Yearbook (2017), when we calculate the descriptive statistical data, the data processing platform uses the Microsoft Excel Software. The descriptive statistical data analysis includes data standardization, entropy weight value, and comprehensive benefit of external environment of ecological compensation for cultivated land protection. Furthermore, when we make the descriptive statistical data presented in space, the Kriging Interpolation Method was used to spatialize the data. Thus, the steps of data processing included four steps. These steps of data processing are listed as follows: 

“the steps of data processing are described as follows. The first step deals with data standardization. In order to eliminate the dimensional effect and the variations in the numerical value of the variable itself. The second step calculates the values of entropy weighting. Step three calculates the values of the comprehensive benefit of external environment of ecological compensation for cultivated land protection. Step four the Kriging Interpolation Method was used to spatialize the data.” (See the Revised Manuscript: from Lines 256-262 of Page 8)

  • Point 10

(1) The term “factors” should be further contextualized here and in the text and headings…

“Table 2. External environment affecting factors of ecological compensation for protecting cultivated land. 298”

Response 10-(1):

-- Thank you very much for the comments.

--As suggested by the reviewer, we have added the contextualized statement about these factors of Table 2. And we have improved the heading of Table 2, which changed “Table 2. External environment affecting factors of ecological compensation for protecting cultivated land” to “Table 2. Factors’ weights of external environment of ecological compensation for cultivated land protect.” (See the Revised Manuscript: from Lines 307-308 of Page 9)

The contextualized statement was added as follows:

“The factors are divided into positive and negative factor categories. When the value of positive external factor is larger, it indicates that the factor will increase the revenue for farmers, who protecting cultivated land. On the contrary, When the value of negative external factor is larger, it shows that the factor will decrease the revenue for farmers, who protecting cultivated land. The weights of factors were calculated by the Equation (6). In Table 2, the weight of the number of service industries above a designated size in districts (cities) and counties (living environment indicators) is a maximum of 0.219. The weight of general public budget expenditures (living environment indicators) and the number of medical institutions (environmental protection indicators) are 0.142 and 0.128. This shows that farmers pay more attention to issues related to their own lives, employment, and security. It is consistent with the desire of farmers in the new era to pursue a high standard of living in addition to meeting their material needs. From the weight of the total agricultural output value (production environment index; at least 0.021), it can be seen that farmers are not paying much attention to their own production environment.” (See the Revised Manuscript: from Lines 294-306 of Page 9)

(2) Revise English, possessive case…

“5.1. Factors weights and comprehensive benefit calculation”

Response 10-(2):

-- Thank you very much for the comments.

--As suggested by the reviewer, we have changed “5.1. Factors weights and comprehensive benefit calculation” to “4.1. Factors weights and comprehensive benefit calculation”. Then, we have changed “4.1. Factors weights and comprehensive benefit calculation” to “4.1. Factors’ weights and comprehensive benefit calculation” (See the Revised Manuscript: in Line 293 of Page 9)

(3) “check all similar language

“obvious” means exactly what?!

“are obvious spatial”

Response 10-(3):

-- Thank you very much for the comments.

--As suggested by the reviewer, we have corrected the unscientific expression about “obvious”.

Terms revision of the “obvious”:

1) “There are obvious differences” in Line 233 of Page 7 has been changed to “There are significant differences”.

2) “obvious and” in Line248 of Page 9 has been removed.

3) “obvious” in Line 314 of Page 9 has been removed.

  • Point 11

Conclusions needs brief contextualization and methodology

Response 11:

-- Thank you very much for the comments.

--As suggested by the reviewer, in order to make more explicit, we have modified the conclusions. Specially, at the beginning and end of this section, we have revised the statement by simplified language. The modified the conclusions is followed (See the Revised Manuscript: from Lines 432‒447 of Page 12):

“The study used GIS to analyze regional differences and assess ecological compensation for urban and rural cultivated land protection at the micro scale. The results show that the willingness of farmers to participate in cultivated land protection is affected by the external environment of cultivated land protection ecological compensation. The comprehensive benefit of external environment for cultivated land protection ecological compensation is the highest at urban-rural junction, which means the comprehensive benefit value is the highest in the “second layer” of Chengdu. The comprehensive benefit shows a Λ trend among the different layers of cultivated land protection ecological compensation in Chengdu. Moreover, in the urban‒rural junction area, the external environment is favorable for cultivated land protection ecological compensation in this area. The external environment can increase farmers’ willingness to protect cultivated land and has a positive effect on farmers’ adoption of cultivated land protection measures. In contrast, in the first and third layer, it has a depressant effect on farmers’ willingness to protect cultivated land.

As the key stakeholders of ecological compensation for cultivated land protection, farmers’ willingness has a significant impact on the implementation of ecological compensation for cultivated land protection, so the government should adopt some incentive measures to increase farmers’ willingness to protect cultivated land.”

  • Point 12

References

“size[J].”?! and many others… “areas [M].”

The reference style is not the one followed by the journal

Pages? “Beijing: Science Press, 2020.”

Response 12:

-- Thank you very much for the comments.

--As suggested by the reviewer, we have modified the references using ACS format. The reference style is the one followed by the journal. The modified the references is followed (See the Revised Manuscript: from Lines 462‒583 of Page 13-15).

Round 3

Reviewer 2 Report

Highlight changes in yellow in a next revision, please. No track changes.

Consider comments in the entire text.

Again…

A study does not use. In this study… was used…!!

“the study used GIS”

Not wants, it is not a person, use “aims” at least…

“this study wants”

The fonts in equations and text differ. Correct…

“disturbance terms. “Bounded rationality” is defined as follows:

(1) 85 (,e, i) Q Lc=

In equation (1), c 86”

Please insert reference equation prior to presenting the equation, and no citaions means complete originality!!

““Bounded rationality” is defined as follows:

(…)”

It des not make sese. To which equation do the references reger to? Only to the second:

“In Equations (2) and (3) [30,43,44],”

But previously only one reference is presented, and it refers to what equations?

“adopt cultivated land protection and the degree of protection [33].”

Equations must be previously introduced (presented…) in the text… as usually done everywhere…

Clarify factos, wat factors…

2.2. Factors selection

Headings must be self-explanatory

Pointed out before: “(1) The term “factors” should be further contextualized here and in the text and headings…”

Check proper spacing… “represents negative ,”

Reference number cannot then include the year too, that is in the references

Chengdu Statistical Yearbook (2017) [46].”

A contextualization to defend the article is necessary in the conclusions section, as present in abstract too:

6. Conclusions 431

The study used GIS to analyze regional differences and assess ecological compensation for urban 432 and rural cultivated land protection at the micro scale.”

I have made extensive comments to the authors, the purpose is only one, to assist the authors in improving the manuscript to be relevant and cited…

Further work is still necessary.

Since I was extensive before I cannot state the same over and over again…

It is the authors’ responsibility to make their best, not achieve yet.

Author Response

We admire that the efforts made by learned reviewer are worth addressing in order to make this work in accordance with good publication standards presenting a quality read. However, we say sorry, if we overlooked or not fully address the concerns, questions or pointers of the learned reviewer. Also, we understand that learned reviewer put a significant effort to identify/highlight the loopholes in this work. We tried our best to improve the manuscript as suggested by the all reviewers.

  • Point 1

Abstract:

(1) A study does not use. In this study… was used…!!

“the study used GIS”

Response 1 - (1):

--Thank you very much for the comments.

--As suggested by the reviewer, we have changed “the study used GIS” with “GIS spatial analysis technology was used”. (See the Revised Manuscript: in Line 16 of Page 1)

(2) “this study wants”

Response 1 - (2):

--Thank you very much for the comments.

--As suggested by the reviewer, we revised “The study wants” with “The study aims” (See the Revised Manuscript: in Line 66 of Page 2)

  • Point 2

(1) The fonts in equations and text differ. Correct…

Response 1 - (1):

--Thank you very much for the comments.

--As suggested by the reviewer, we have revised the fonts of equations with the “Palatino Linotype” font. That’s consistent with the text font. (See the Revised Manuscript: in Equations (1)-(7))

(2) “disturbance terms. “Bounded rationality” is defined as follows:

(1) 85 (,e, i) Q Lc=

In equation (1), c 86”

Please insert reference equation prior to presenting the equation, and no citaions means complete originality!!

““Bounded rationality” is defined as follows:

(…)”

Response 1 - (2):

--Thank you very much for the comments.

--Before Equation (1), the description of the equation has been added according to the cited references [30‒32], including the meaning of the equation, the applicability of references, and the meaning of the parameters in the equation. As suggested by the reviewer, revised portions are highlighted using the "Yellow" function in Microsoft Word. (See the Revised Manuscript: from Lines 81-89 of Page 2)

The revised portions of this manuscript are listed as follows:

“The Bounded Rationality Theory (BRT) illustrates the process of decision making and indicates various elements of decision making. The rationality of decision-making under BRT is not the absolute optimal solution, but is the most satisfactory solution among all alternative solutions [30‒32]. According to BRT, the decision-making object is influenced by its own cognitive factors, external environmental factors, and random interference terms. The given expression further clarifies the composition factors of BRT [30,31]:

                                   (1)

In Equation (1), c represents the cognitive factors of the decision-making subject, e represents the external environment factors, and i represents the random interference terms.”

  • Point 3

(1) It des not make sese. To which equation do the references reger to? Only to the second:

“In Equations (2) and (3) [30,43,44],”

But previously only one reference is presented, and it refers to what equations?

Equations must be previously introduced (presented…) in the text… as usually done everywhere…

Response 3-(1):

--We thank the reviewer for pointing out these deficiencies.

--The Equations (2) is based on authors own illustration drawn from the studied empirical and theoretical literature. However, Equation (3) is based on prior literature [30,43,44] (See the Revised Manuscript: from Lines 237-242 of Page 7). To make it more clearly, we have added the detail description of the equation based on PSBA. (See the Revised Manuscript: from Lines 223-234 of Page 6).

The revised portions of this manuscript are listed as follows:

After a comprehensive assessment of the external environment, farmers will consider whether to adopt measures to protect the cultivated land. Then, for farmers who take measures to protect cultivated land, they are most concerned about how to maximize their profits [33]. Based on the PSBA, under the current external environment of the region, the predictable total revenue and the total cost are the key factors for famers to consider. When the predictable total revenue is greater than the total cost, the comprehensive benefit of external environment of ecological compensation for cultivated land protection is positive. This means that the external environment of the region is conducive to farmers’ increasing willingness to protect cultivated land. However, when the predictable total revenue is less than the total cost, the comprehensive benefit of external environment of ecological compensation for cultivated land protection is negative. This means that the external environment of the region is not conducive to improving farmers’ willingness to protect cultivated land. Therefore, the model specification can be showed as:

                                              (2)

                                              (3)

In Equations (2) – this equation is based on authors own illustration drawn from the studied empirical and theoretical literature – and, however Equation (3) is based on prior literature [30,43,44]. In Equation (2), B represents the comprehensive benefit of external environment of ecological compensation for cultivated land protection, TR refers to the predictable total revenue of farmer households under the external environment for ecological compensation of cultivated land protection, TC refers to the predictable total cost of farmer households. Likewise, in Equation (3), Pi represents some positive indicators, n is the number of positive indicators, Ci represents negative, and Ki represents the weight of indicators, m is the number of negative indicators.

(2) “adopt cultivated land protection and the degree of protection [33].”

Response 3-(2):

--We thank the reviewer for pointing out these deficiencies.

--We have revised “farmers will consider whether to adopt measures to protect the cultivated land and degree of protection [33].” to “farmers will consider whether to adopt measures to protect the cultivated land. In particular, for farmers who take measures to protect cultivated land, they are most concerned about how to maximize their profits [33].” (See the Revised Manuscript: from Lines 224-225 of Page 6). As suggested by the reviewer, we have checked and improved the similar issues in this manuscript.

Terms of revised the similar issues are followed:

1) We have changed “Smith combined space cost curve theory by Weber and space revenue curve theory by Losch [43,44],” to “Smith combined space cost curve theory by Weber [43] and space revenue curve theory by Losch [44],”. (See the Revised Manuscript: from Lines 186-187 of Page 5).

2) We have changed “However, due to the impact of market supply and demand prices [53,54],” to “However, due to the impact of market supply [53] and demand prices [54],”. (See the Revised Manuscript: from Lines 460-461 of Page 13).

  • Point 4

(1) Clarify factos, wat factors…

“2.2. Factors selection”

Headings must be self-explanatory

Pointed out before: “(1) The term “factors” should be further contextualized here and in the text and headings…”

Response 4 - (1):

--Thank you very much for the comments.

--As suggested by the reviewer, to make the heading more self-explanatory, we have changed the heading of “2.2. Factors selection” to “2.2. External environment factors of cultivated land ecological compensation”.

In addition, we have added the contextualized description of factors at the beginning of Section 2.2. The contextualized description includes the importance of cultivated land for human beings, especially, for farmers, the relationship between cultivated land and urbanization of China, and the external environment factors selection of cultivated land ecological compensation. (See the Revised Manuscript: from Lines 118-139 of Pages 3-4).

The revised portions are listed as follows:

“Cultivated land is the absolute foundation of human beings’ survival today and in the future and, as such, it must be protected from conversion into built‒up environments and fragmentation. With the largest population in the world and rapid urbanization, China has a particular need to protect its cultivated land [2]. Since the start of the reforms and openness policy in 1978, urbanization of China has been accompanied by a series of large‒scale land use changes [20]. Meanwhile, there are stark differences between urban and rural areas in China [1‒3,27]. From the perspective of geographical location, farmers can be divided into urban villages, rural areas near the urban areas, and remote rural areas. Based on the BRT, the willingness of farmers to undertake cultivated land protection is affected by the external environment [36]. The yield of cultivated land is closely related to the factors agricultural production environment. The improvement of agricultural production environment contributes to the increase of cultivated land yield. However, the regional agricultural production environment depends on the cultivated land protection policy environment [22]. To the farmers, the degree of protection for cultivated land is influenced by the different factors of the living environments. And the employment environment affects the number of farmers working in agriculture. Furthermore, in China, cultivated land serves as both a means of production and as social insurance for farmers. The agricultural subsidy and ecological compensation for cultivated land can directly increase farmers’ income and consumption level [17]. The representative variables were analyzed in this study. The variables affecting ecological compensation for farmers’ cultivated land protection were defined from the perspective of the agricultural production environment, farmers’ living environment, farmers’ employment environment, farmers’ safeguarding environment, the market environment, and policy environment of the ecological compensation for cultivated land protection t (Figure 2).”

(2) Check proper spacing… “represents negative ,”

Response 4 - (2):

--Thank you very much for the comments.

--As suggested by the reviewer, we have revised “represents negative ,” with “represents negative,”. (See the Revised Manuscript: in Line 243 of Pages 7).

  • Point 5

Reference number cannot then include the year too, that is in the references

“Chengdu Statistical Yearbook (2017) [46].”

Response 5:

-- Thank you very much for the comments

-- In this manuscript, the “Chengdu Statistical Yearbook (2017) is the full name of literature, and the ”2017” and “Chengdu Statistical Yearbook” shouldn’t be separated. To make it more clearly, we have changed “Chengdu Statistical Yearbook (2017) [46]” to “Chengdu Statistical Yearbook 2017 [46]”. (See the Revised Manuscript: in Line 283 of Page 8)

  • Point 6

(1) A contextualization to defend the article is necessary in the conclusions section, as present in abstract too:

Response 6-(1):

--We thank the reviewer for pointing out these deficiencies.

--As suggested by the reviewer, we have added the brief contextualization at the beginning of the conclusions.

The added the brief contextualization is listed as follows:

“As the key stakeholders of ecological compensation for cultivated land protection, farmers’ attitudes, and especially their willingness, are key variables that help determine the effectiveness of polices that seek to coordinate urban‒rural development and food security.” (See the Revised Manuscript: from Lines 467-469 of Page 13.)

(2) “6. Conclusions 431

The study used GIS to analyze regional differences and assess ecological compensation for urban 432 and rural cultivated land protection at the micro scale.”

Response 6-(2):

--We thank the reviewer for pointing out these deficiencies.

--As suggested by the reviewer, we have corrected “The study used GIS to analyze regional differences and assess ecological compensation for urban and rural cultivated land protection at the micro scale.” with “In this study, GIS spatial analysis technology was used to analyze regional differences and assess ecological compensation for urban and rural cultivated land protection at the micro scale.” (See the Revised Manuscript: from Lines 469-471 of Page 13.)

  • Point 7

In order not to overlooked and fully address the concerns, questions or pointers of the learned reviewer, we have combed through the reviewers' opinions, including previous ones.

We have added the introduced description and references for all equations in the text. Corresponding, we changed the number of references in this manuscript.  

Response 7:

-- Thank you very much for the comments

In Equation (5), the added the introduced description is listed as follows:

“Based on the standardized data, the information entropy and entropy weigh of each index can be defined. Information entropy is related to all possibilities. Every possible event has a probability. Information entropy is the average amount of information we get when an event happens [47]. Thus, mathematically, entropy is the expectation of the amount of information.” (See the Revised Manuscript: from Lines 300-303 of Page 9.)

In Equation (6), the added the introduced description is listed as follows:

“Entropy–weight method is used in this study. This method is objective when the index is weighted. According to the variation degree of each index, the entropy–weight method calculates the entropy weight of each index by using information entropy [47]. Then, to make the entropy weight more objective, the weight of each index is modified by the entropy weight.” (See the Revised Manuscript: from Lines 309-312 of Page 9.)

In Equation (7), the added the introduced description is listed as follows:

“According to authors own illustration drawn from the studied empirical and theoretical literature, the results of the comprehensive level of factors comes from the sum of entropy weight multiplied by standardized data. This means that the comprehensive level of the positive and negative factors in the external environmental of ecological compensation for cultivated land protection can be calculated based on the standardized data and entropy weight of each index. Therefore, the comprehensive level of factors can be computed in the following equation:” (See the Revised Manuscript: from Lines 308-323 of Page 9.). ”In Equation (7), which is based on authors own illustration drawn from the studied empirical and theoretical literature.” (See the Revised Manuscript: from Lines 325-326 of Page 9.).

We again thank the reviewer for very helpful suggestions throughout the manuscript.
